# Optogenetic feedback control of neural activity

Jonathan P Newman[1,2]*, Ming-fai Fong[1,2,4], Daniel C Millard[2,3], Clarissa J Whitmire[2], Garrett B Stanley[2], Steve M Potter[2]*

[1]Picower Institute for Learning and Memory, Department of Brain and Cognitive Sciences, Massachusetts Institute of Technology, Cambridge, United States; [2]Laboratory for Neuroengineering, Department of Biomedical Engineering, Georgia Institute of Technology, Atlanta, United States; [3]Axion Biosystems, Atlanta, United States; [4]Department of Physiology, Emory University School of Medicine, Atlanta, United States

**Abstract** Optogenetic techniques enable precise excitation and inhibition of firing in specified neuronal populations and artifact-free recording of firing activity. Several studies have suggested that optical stimulation provides the precision and dynamic range requisite for closed-loop neuronal control, but no approach yet permits feedback control of neuronal firing. Here we present the 'optoclamp', a feedback control technology that provides continuous, real-time adjustments of bidirectional optical stimulation in order to lock spiking activity at specified targets over timescales ranging from seconds to days. We demonstrate how this system can be used to decouple neuronal firing levels from ongoing changes in network excitability due to multi-hour periods of glutamatergic or GABAergic neurotransmission blockade in vitro as well as impinging vibrissal sensory drive in vivo. This technology enables continuous, precise optical control of firing in neuronal populations in order to disentangle causally related variables of circuit activation in a physiologically and ethologically relevant manner.

*For correspondence:
jpnewman@mit.edu (JPN); steve.
potter@bme.gatech.edu (SMP)

**Competing interests:** The authors declare that no competing interests exist.

**Reviewing editor**: Marlene Bartos, Albert-Ludwigs-Universität Freiburg, Germany

## Introduction

Feedback is essential for controlling complicated systems. It can be used to define system dynamics and decouple causal interactions. Recently, a diverse set of specialized techniques that employ elements of feedback control have emerged for studying adaptation in neuronal micro-circuits (*Ahrens et al., 2012*), using electrical stimulation to control spike latency (*Wallach et al., 2011*) and firing levels (*Wagenaar et al., 2005*; *Newman et al., 2013*), improving brain-computer interfaces (*Velliste et al., 2008*; *Cunningham et al., 2011*), inducing motor plasticity (*Jackson et al., 2006*), and controlling intracellular firing rate (*Miranda-Domínguez et al., 2010*). True feedback control has been most broadly applied in neuroscience research using the voltage clamp, which is used to decouple the membrane potential from causally related voltage-dependent conductances. This approach has provided the foundation for our understanding intracellular electrochemical signaling and demands extension to other features of neural activity.

The neuronal firing rate is a basic feature of codes for motor action (*Georgopoulos et al., 1988*), vision (*Steinmetz et al., 1987*), and place (*Zhang et al., 1998*). Changes in thalamic firing tone can alter cortical receptive fields (*Stoelzel et al., 2009*) and the nature of temporal coding within the thalamocortical pathways (*Wang et al., 2010*). Long-term changes in network firing levels can trigger a multitude of homeostatic processes that regulate circuit excitability and stability (*Turrigiano et al., 1998*; *Corner et al., 2002*; *Turrigiano, 2011*). A system analogous to the voltage clamp, but capable of precise, bidirectional control of circuit firing levels could be used to identify the independent role of

**eLife digest** Cells called neurons use electrical signals to rapidly carry information around the body. When a neuron is activated, it generates (or 'fires') a short electrical impulse that travels along the cell to relay a message to other neurons, muscles or organs. Optogenetics is a technique that allows scientists to genetically modify neurons to produce proteins that make them light sensitive.

One of the most commonly used light-sensitive proteins is called channelrhodopsin-2. It is activated by blue light and increases the electrical activity of neurons. Another protein is called halorhodopsin, which responds to yellow light and inhibits the firing of neurons. By shining light of particular colors onto neurons that produce these and other light-sensitive proteins, it is possible to manipulate the activity of large populations of neurons.

Most previous optogenetic experiments have involved altering the activity of neurons and then observing the outcome at a later point in time. However, it would be very useful to be able to alter the amount of optical stimulation to achieve particular levels of neuron activity in real time. To achieve this, the level of neuron activity at any point in time would need to be quickly compared to the desired level, so that optogenetics could be used to increase or decrease the firing of neurons as appropriate.

Newman et al. have now developed an optogenetic system called 'optoclamp' that can control the activity of neurons in real time. In neurons grown in cell culture, the optoclamp is able to hold the level of neuron activity at particular values for periods of time ranging from 60 seconds to 24 hours. It can be used to restore and maintain the baseline level of neuron activity in the presence of drugs that would otherwise produce large increases or decreases in the firing of neurons. Moreover, in anaesthetized rats, the optoclamp can prevent some neurons from being activated even when the rats' whiskers move, which would normally change their firing level.

Newman et al.'s findings open the door to a new type of neuroscience experiment where it is possible to manipulate activity patterns as they are produced by the brain. This will help researchers to understand how particular patterns of brain activity are linked to learning, memory, and behavior.

firing rate in downstream processes in spite of changes to causally-related variables. For instance, long-term changes in population firing have long been thought to initiate compensatory homeostatic mechanisms, but a causal link has remained elusive. Direct bidirectional control over population firing rates would allow us to test this hypothesis directly. Optogenetic tools are routinely used to provide genetically specified, millisecond time-scale stimulation or suppression of neural activity with light (*Mattis et al., 2011*) during simultaneous, artifact-free electrical recording. The ability to simultaneously perturb and measure neural activity form the basic elements of a feedback loop, which can be exploited to control firing. Although several studies have presented closed-loop optogenetic stimulation techniques (*Leifer et al., 2011*; *Stirman et al., 2011*; *Paz et al., 2012*; *Krook-Magnuson et al., 2013*; *O'Connor et al., 2013*; *Siegle and Wilson, 2014*), no approach yet permits feedback control of neuronal firing levels.

Here we describe and quantify optogenetic feedback control ('optoclamping'), a method enabling continuous, bi-directional, closed-loop firing rate control both in vitro and in vivo. We show that the optoclamp allows precise control of population firing levels in dissociated cortical networks over timescales ranging from seconds to days. We characterize the effects of different control schemes, algorithm parameters, and optical waveforms on the precision of feedback control and higher-order statistics of population activity. We show that firing rate control can be achieved over many hours and used to restore pre-drug firing levels during chronic blockade of excitatory or inhibitory synaptic transmission. Using this approach, we decouple the effects of suppressed neurotransmission from the indirect effects on network firing, and find that changes in firing levels are not required to induce homeostatic alterations in network excitability. Finally, we show how optogenetic feedback can be used to control firing activity in vibrissal somatosensory thalamus of rats. We find that background thalamic activity levels can be controlled during ongoing sensory input without corrupting the fine-scale temporal structure of whisker-evoked spike trains. Together, our results demonstrate that the optoclamp is an effective general tool for decoupling neural firing from other variables that would normally affect network excitability.

## Results

### Characterizing bidirectional optical control signals

To characterize the range of evoked firing levels that could be achieved using multimodal optical stimulation in dissociated cortical networks, we stimulated excitatory cells expressing channelrhodopsin-2(H134R) ($ChR2_R$) (*Nagel et al., 2005*) and enhanced halorhodopsin-3.0 (eNpHR3.0) (*Gradinaru et al., 2008*) while recording spiking activity using a 59-channel microelectrode array (MEA; *Figure 1A*) (*Newman et al., 2013*). For $ChR2_R$ activation, a single dimensionless excitatory control variable, $U_C$, simultaneously modulated the pulse width, frequency, and intensity of homogeneous 465 nm stimuli (*Equations 7–9*, 'Materials and methods'; *Figure 1—figure supplement 1* and *Figure 1—figure supplement 2*). For eNpHR3.0 activation, we defined a second control variable, $U_H$, proportional to the continuous intensity of a 590 nm LED (*Equation 10*). We applied $U_C$ and $U_H$ ranging from 0 to 1, for randomly interleaved, 60-s stimulation epochs (2 cultures, 50 trials/culture). Evoked population firing rates were positively correlated with $U_C$ and negatively correlated with $U_H$ (*Figure 1B*). $U_C$-evoked firing levels saturated at approximately 12.5 Hz/unit corresponding to $U_C = 0.47$ (freq. = 14.7 Hz, pulse-width: 2.4 ms, power at 465 nm: 6.9 mW·mm$^{-2}$). Firing rate suppression saturated at 0.04 Hz/unit corresponding to $U_H = 0.15$ (power at 590 nm: 1.8 mW mm$^{-2}$). The monotonic relationships between $U_C$ and $U_H$ and network firing levels indicated their applicability as closed-loop control signals.

We also tested the robustness of $U_C$ and $U_H$ for modulating firing levels during blockade of AMPAergic, NMDAergic, or GABAergic transmission using CNQX, AP5, or bicuculline, respectively ('Materials and methods'). Synaptic blockade strongly affected mean spontaneous firing levels (CNQX: −74.2%, AP5: −66.6%, bicuculine: +357%) and network-level signal propagation (*Figure 1C*). In spite of this, firing in CNQX- and AP5-treated networks could be driven over the same dynamic range as the drug free condition (*Figure 1B*; CNQX: 0.057 to 13.3 Hz/unit, AP5: 0.088 to 12.6 Hz/unit), indicating that optogenetic input could compensate for depressed excitatory transmission. Bicuculline greatly reduced the dynamic range of evoked network activity indicating a loss of reliable activity modulation (*Figure 1B*; 0.095 to 5.1 Hz/unit).

Although time-averaged firing levels were monotonically related to $U_C$ and $U_H$ (*Figure 1B*), open-loop stimuli lost effectiveness throughout each 60-s trial causing significant second-to-second drift in evoked activity levels (*Figure 1D,E*). This effect was consistent across synaptic blocker conditions. Decreases in stimulus efficacy using $ChR2_R$ were likely due to network adaptation, rather than changes in $ChR2_R$-mediated photocurrents (*Mattis et al., 2011*). Aside from network adaptation, deceased efficacy of eNpHR3.0-mediated firing suppression likely resulted from a loss of synaptic inhibition due to intracellular Cl$^-$ accumulation (*Raimondo et al., 2012*) and decreased outward photocurrents due to pump desensitization (*Mattis et al., 2011*).

### Proportional-integral control of network firing

We developed a proportional-integral (PI) feedback controller to clamp network population activity in dissociated cortical networks in the face of uncontrolled fluctuations in neuronal excitability and opsin dynamics (*Equations 1–6*). The PI algorithm updated $U_C$ and $U_H$ in real-time in order to minimize the difference ('error') between the measured network firing rate and a target level ('Materials and methods'). We tested the controller using 60-s, randomly-ordered targets ranging from 0–10 Hz/unit (*Figure 2A*; 7 cultures, 11 trials/culture), and successful control was achieved in over 90% of trials (*Figure 2B*; 71/77 trials; mean RMS tracking error 0.14 ± 0.091 Hz/unit). Tracking error increased with target rate, and occasionally the stimulator saturated before the trial was complete (e.g., *Figure 2A*, grey line). Control settling time varied across preparations and was not correlated with the target firing rate (mean ± SD, 7.83 ± 6.07 s; *Figure 2—figure supplement 1*). Although we generally used discrete steps in the target firing rate, the controller was also capable of tracking continuously varying targets (*Figure 2—figure supplement 2*).

We explored the parametic sensitivity of the controller by changing the value of either the proportional gain ($K$), the integral-error time constant ($T_i$), or the firing rate filter time constant ($\tau$), while the remaining two parameters were held at their nominal values ($K = 0.1$, $T_i = 1$ s, $\tau = 2.5$ s). Functional control was achieved at $K < 1.0$, 1.0 s $< T_i <$10 s, and 0.5 s $<\tau <$10 s. Outside of these bounds, closed-loop dynamics were unstable and/or firing levels exhibited significant target offsets (*Figure 2—figure supplement 3* through *Figure 2—figure supplement 5*).

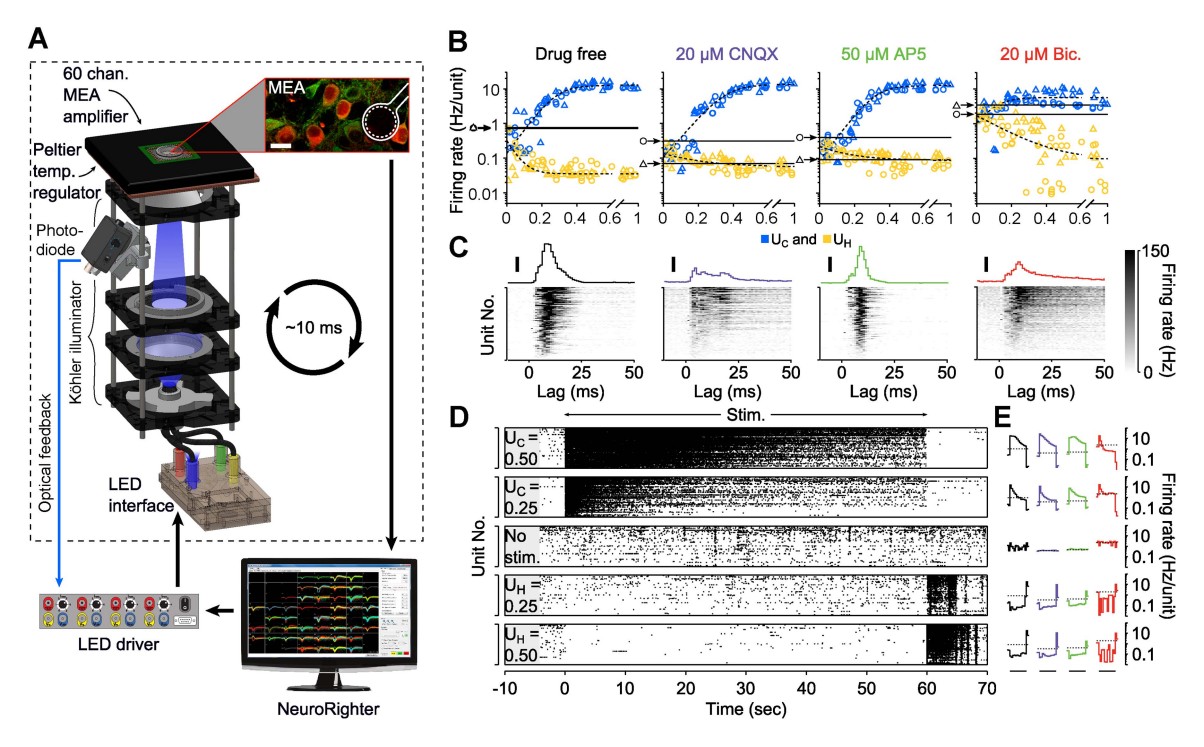

**Figure 1**. Optogenetic modulation of network activity in vitro. (**A**) Multichannel recording, processing, and stimulation system. A 59-channel amplifier detects spiking activity produced by cells close to electrodes (white outline). Neurons express ChR2$_R$-mCherry (red) and eNpHR3.0 under the CaMKllα promoter (green: immunoreactivity for CaMKllα; scalebar: 20 μm). Electrode voltages are processed in real-time and can be used to update an LED stimulator feeding a homogeneous Köhler illuminator below the MEA. An optical feedback circuit (blue line) ensures distortion free blue stimulus waveforms. (**B**) Time-averaged firing rates of two cultures (△ and ○) in response to 60-s applications of randomly valued $U_C$ and $U_H$ during different forms of synaptic blockade. Black horizontal bars indicate the cultures' spontaneous firing levels. Blue and yellow symbols indicate the mean firing level over a single trial at the corresponding value of $U_C$ and $U_H$, respectively. The dotted lines are least-squares fits used to estimate the $U_C$ and $U_H$ saturation points provided in the text. (**C**) PSTH of individual units (grey scale) and the unit-averaged PTSH in response to 1 millisecond 5 mW mm$^{-2}$ blue light pulses for each drug condition. Scale bars, 50 Hz/unit. (**D**) Raster plots for 87 detected units during 60-s applications of $U_C$ and $U_H$. The firing rate evoked by stimulation using a particular value of $U_C$ and $U_H$ decays over the course of the protocol. (**E**) The trial-averaged firing rate profiles for the stimulus levels presented in (**D**) across drug conditions. Black horizontal lines indicate the 60 s stimulation period. Dotted lines indicate spontaneous firing levels. Note the log scale. In (**C**) and (**E**), line colors indicate the drug conditions above each panel in (**B**).

The following figure supplements are available for figure 1:

**Figure supplement 1**. Optical characteristics of the in vitro stimulator and in vivo fiber.

**Figure supplement 2**. Expression time course of AAV2-CaMKIIα-ChR2(H134R)-mCherry.

The control signals, $U_C$ and $U_H$, were highly variable across networks, even for the same target rate (*Figure 2C*). This variability likely reflects heterogeneous network excitability, opsin expression, synaptic connectivity, and developmental processes (*Wagenaar et al., 2006*) and suggests that the controller continuously adapts to ongoing changes in network excitability in order to precisely clamp firing levels. To test this possibility, we delivered open-loop replay of successful closed-loop control signals and found they were incapable of controlling firing (Figure 4, *Figure 2* cultures). This demonstrates the necessity of real-time feedback to achieve precise control of neural firing, even for same target rate within single preparations.

Next, we used the PI controller to clamp network firing levels to targets between 0 and 10 Hz/unit during blockade of AMPAergic and NMDAergic transmission (*Figure 2—figure supplement 6A*; 'Materials and methods'). Control performance was equivalent to the drug-free case and control failure was isolated to high target rates (9–10 Hz/unit) due to stimulus saturation (*Figure 2—figure supplement 6B–E*). We also tested PI control during blockade of GABAergic transmission and found

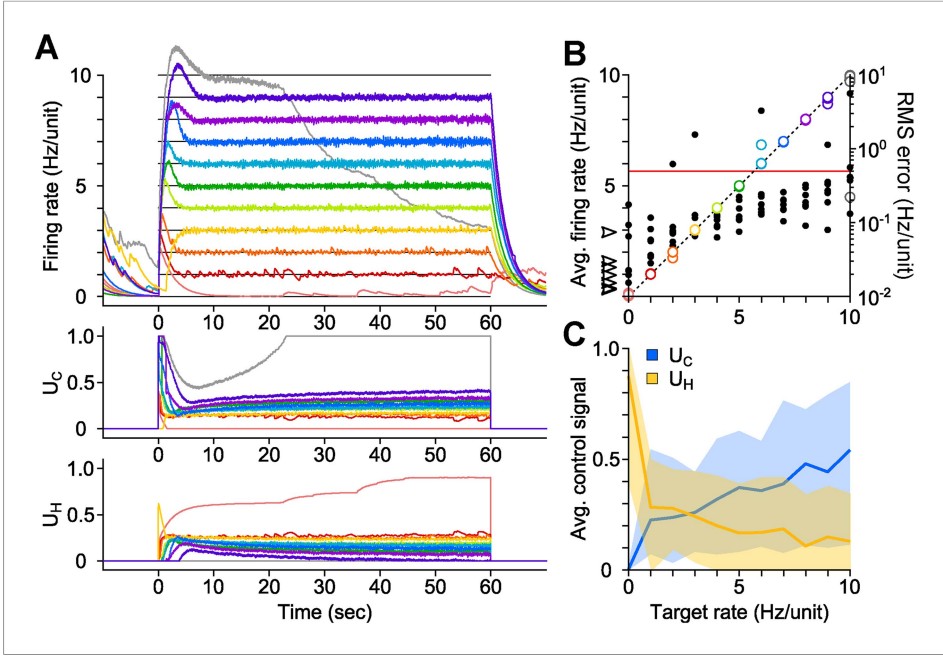

**Figure 2**. PI optical feedback allows precise control of network firing levels over 1-min epochs. (**A**) (*Top*) Network firing rate during different trials (colors). Target firing levels (black lines) ranged from 0 to 10 Hz and were applied in random order. (*Bottom*) Control signals, $U_C$ and $U_H$, required during closed-loop control. For this network, the controller saturated while attempting to clamp network firing at 10 Hz/unit, resulting in a control failure (grey trace). (**B**) Time-averaged firing rates for seven different networks during PI control (left axis, colors). The dotted line is identity representing perfect closed-loop control. The spontaneous firing rates of each network are indicated by black arrows. The RMS error between the measured and target firing for each network is shown as a function of the target rate (right axis, black markers). A trial was considered successful if the RMS error between the target and achieved firing rate was less than 0.5 Hz/unit (red line). (**C**) Time- and culture-averaged successful control signals vs target firing rates. The shaded areas indicate the minimum and maximum value across networks. All temporal averages in this figure were taken over the final 30 s of the control epoch.

The following figure supplements are available for figure 2:

**Figure supplement 1**. PI settling time in vitro.

**Figure supplement 2**. PI feedback permits control during a continuously changing target rate.

**Figure supplement 3**. Effects of proportional gain (*K*) on closed-loop stability of PI control in vitro.

**Figure supplement 4**. Effects of integral time-constant ($T_i$) on closed-loop stability and accuracy of PI control in vitro.

**Figure supplement 5**. Effects of the firing-rate estimation time-constant ($\tau$) on closed-loop stability of PI control in vitro.

**Figure supplement 6**. PI control of firing levels during synaptic blockade in vitro.

that reliable control was not possible (*Figure 2—figure supplement 6*). This is likely due to the destabilizing effects of bicuculline, which caused the controller to oscillate. This result is consistent with studies indicating a general role of reduced inhibition in diseases of circuit instability such as temporal lobe epilepsy (*Kobayashi and Buckmaster, 2003*) and Dravet syndrome (*Dutton et al., 2013*).

To demonstrate PI control over more extended time periods, we clamped firing at a set of randomly selected, 5-min long target firing levels which switched without downtime (50 min. total clamp time; *Figure 3*). During each 5-min step, the controller made rapid, second-to-second adjustments in stimulus

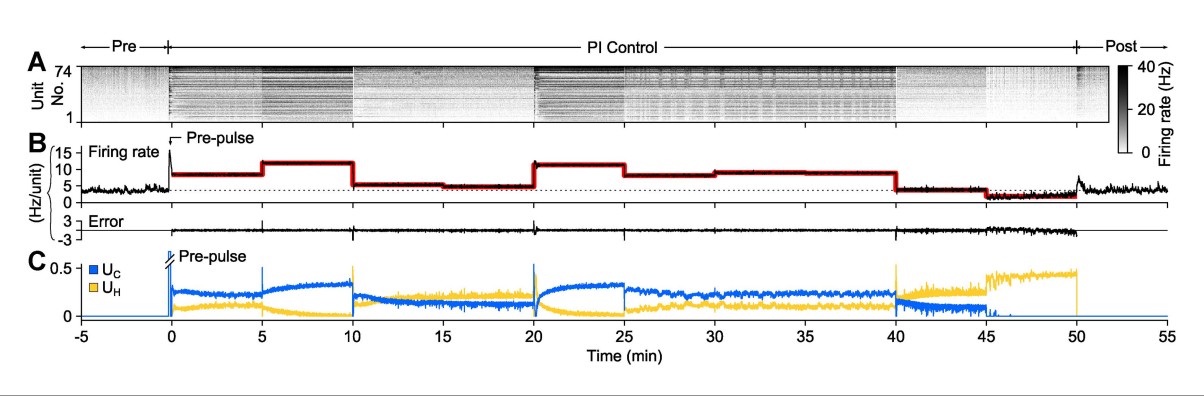

**Figure 3**. PI feedback control to track a changing target rate. (**A**) Firing rate of detected units. Each row displays the firing rate of a particular unit, encoded by the grey-scale to the right (1 s bins). (**B**) The average firing rate of the network (black), the target firing rate (red), and the error signal during different control periods. The pre-control firing rate is indicated by a dotted line. (**C**) Optical control signals delivered by the PI controller during the control epoch.

intensity to maintain the instantaneous target rate, while slower changes in stimulus intensity occurred over minutes. Minute-to-minute changes in the control signal intensity likely reflect short-term synaptic depression and changes in cellular excitability that accrued over each control epoch. Therefore, the control signal can be used as a readout of network excitability, analogous to how injected current from a voltage clamp amplifier can be used to asses cellular excitability.

To test how different stimulus waveforms affected network response correlations during PI control, we mapped $U_C$ onto triangular, sinusoidal, pseudorandom binary sequence (PRBS), and direct intensity modulation inputs (*Tchumatchenko et al., 2013*). Each permitted successful closed-loop control, but notably, the choice of stimulus waveform significantly affected the peak firing correlation ($P = 6.5 \times 10^{-24}$) and synchrony ($P = 10^{-22}$) of the population response (*Figure 4*). The square pulse trains typically used for ChR2-based stimulation (*Mattis et al., 2011*) resulted in periodic, highly correlated population firing (*Figure 4A*). Compared to square pulses, sinusoidal stimuli decreased peak unit-to-unit firing correlations (−10.6%, p = 0.028), but did not affect synchrony. PRBS stimuli reduced peak correlations (−24.0%, p = 6.8 × 10⁻¹³) and firing synchrony (−17.9%, p = 4.6 × 10⁻⁸). Continuous light modulation increased both correlations (+21.3%, p = 4.3 × 10⁻⁵) and synchrony (+41.4%, p = 2.5 × 10⁻⁹). Triangular pulses did not affect correlation or synchrony. Importantly, while periodic input signals produced a periodic response, PBRS input and continuous intensity modulation resulted in non-periodic firing. Therefore, altering the temporal characteristics of excitatory stimulus waveforms resulted in remarkably different higher-order firing statistics while still enabling successful PI control. This emphasizes the fact that, in its current form, the optoclamp only controls population firing levels and leaves more complex features of neural activity unconstrained and subject to the influence of network connectivity, network dynamics, and the nature of the stimulus signal.

## Multi-hour control of firing rates

Planar MEAs allow stable extracellular recordings over long timescales. Previous studies have used MEAs to monitor spiking activity over many hours and correlated recorded activity with homeostatic or developmental changes in network properties (*Wagenaar et al., 2006*; *Minerbi et al., 2009*). Compared to simply measuring spiking activity, controlling mean firing rates over long timescales would enable investigations of causal, rather than correlative, relationships between spiking and long-term, activity-dependent processes.

To this end, we developed an 'on-off' controller to clamp network activity across many hours. To clamp firing rates above spontaneous levels, this controller delivered a blue light pulse (5 ms, 13.4 mW mm⁻², 465 nm) when the integral error exceeded zero (*Equation 12*). We tested on-off control by clamping network firing rates within a single culture to seven elevated setpoints (ranging from 0.75 to 6 Hz/unit) for 12-hr epochs (*Figure 5*). The on-off controller successfully held

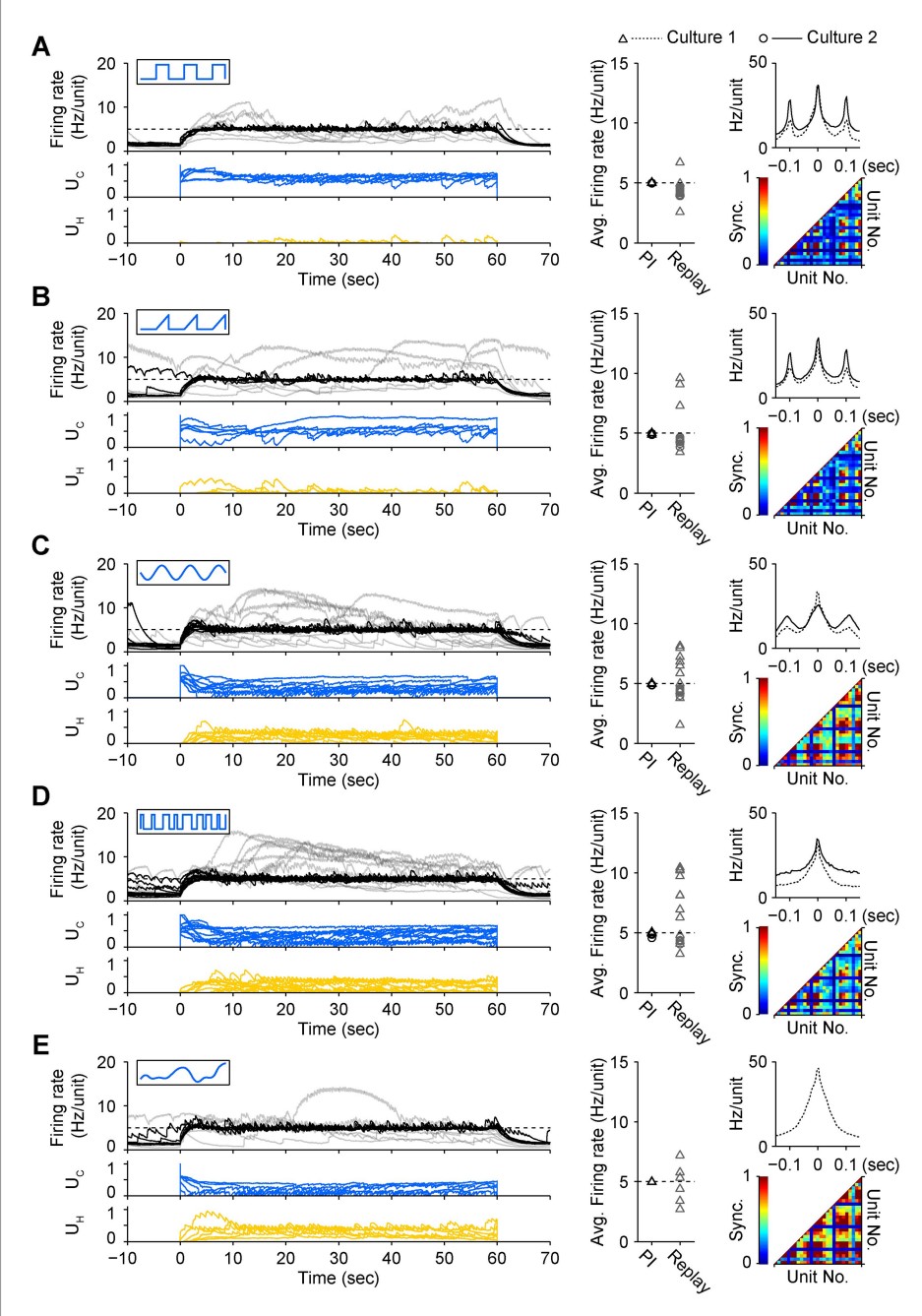

**Figure 4**. A diverse set of optical input signals can be used to clamp network firing rate, and accurate firing rate control requires closed-loop stimulation. (**A**) *Left*, network firing rates during closed-loop control (black) and during replay of closed-loop control signals in open-loop (grey), along with corresponding control inputs $U_C$ (blue) and $U_H$ (yellow). These time series data were derived from culture 1. Note that in all instances, open-loop replay of input signals recorded from previous successful closed-loop control failed to clamp firing levels and resulted in erratic activity levels over the control epoch. *Middle*, time-averaged firing rates for both cultures during closed-loop control (black) and during subsequent replay of control signals in open-loop (grey). *Right*, average unit-to-unit cross-correlogram for both cultures (top, bin = 5 ms) and unit-to-unit synchronization structure for culture 1 (bottom) during optogenetic feedback control. Synchronization was defined as,

$$\text{Sync}_{i,j} = \frac{N_{cc}}{\sqrt{\left(N_i^2 + N_j^2\right)\big/2}},$$

*Figure 4. continued on next page*

*Figure 4. Continued*

where $N_{cc}$ is number of correlated events within ± 10 ms, and $N_i$ and $N_j$ are the number of spikes from units $i$ and $j$ used to calculated the cross-correlogram.

(**B**) Same as (**A**) for triangle optical stimuli modulated according to.

$$\text{Pulse freq.}_{465\text{ nm}} = 10\text{ Hz},$$

$$\text{Rising slope.}_{465\text{ nm}} = 0.22\ \frac{\text{mW}}{\text{ms} \cdot \text{mm}^2},$$

$$\text{Peak power}_{465\text{ nm}} = 13.4U_C\ \frac{\text{mW}}{\text{mm}^2}.$$

(**C**) Same as (**A**) for sinusoidal optical stimuli modulated according to.

$$\text{Optical power}_{465\text{ nm}} = 13.4U_C\sin(2\pi10t)\ \frac{\text{mW}}{\text{mm}^2}.$$

(**D**) Same as (**A**) for pseudo-random binary sequence of optical pulses modulated according to.

$$\text{Update freq.}_{465\text{ nm}} = 150\text{ Hz},$$

$$\text{Peak power}_{465\text{ nm}} = 13.4U_C\ \frac{\text{mW}}{\text{mm}^2}.$$

(**E**) Same as (**A**) for continuous optical stimuli modulated according to.

$$\text{Optical power}_{465\text{ nm}} = 13.4U_C\ \frac{\text{mW}}{\text{mm}^2}.$$

Each protocol was performed in the same culture.
Periodic stimuli (panels **A**–**C**) were applied at 10 Hz so that the periodicity of evoked activity would be apparent in the correlation functions. In all cases, the 590 nm light was modulated according the standard control scheme (*Equation 10* of 'Materials and methods'). Note that each input type evokes a unique correlation and synchronization structure while still achieving accurate firing rate control.

firing rates at six of these targets, saturating only at 6 Hz/unit after ~7 hr (*Figure 5A,B*). Notably, the stimulation frequency required to maintain each target firing rate was better correlated to the difference between the target and the pre-clamp firing rate than the target rate alone ($R^2 = 0.63$ vs 0.52; *Figure 5—figure supplement 1*). This indicates that alterations in network excitability across different experimental days were reflected in the intensity of the control inputs (*Figure 5C,D*; *Figure 5—figure supplement 1A*) and suggests that optogenetic feedback control can be used to study changes in neuronal network dynamics over developmental timescales.

To clamp firing rates below spontaneous levels, the on-off controller delivered continuous yellow light (~11 mW mm$^{-2}$, 590 nm) when the integral error signal fell below zero (*Equation 13*; 'Materials and methods'). Because chronic activation of eNpHR-3.0 produced relatively weak photocurrents and induced a depolarizing shift in the chloride reversal potential (*Raimondo et al., 2012*), we found that it was inadequate for long-term control. For this reason, archaerhodopsin-3.0 (Arch-3.0) was used for multi-hour firing rate suppression (*Chow et al., 2010*; *Mattis et al., 2011*) ('Materials and methods'). To validate this strategy, we clamped a culture's spiking activity to ~60% of its spontaneous firing rate (from 1.23 to 0.75 Hz/unit) over a 3-hr epoch (*Figure 5—figure supplement 2*).

The on-off and PI control schemes have distinct advantages and disadvantages. PI control provided rapid response times and low RMS error over small time windows, but imposed strong, short timescale (~50 ms) response correlations between units (*Figure 5C*; *Figure 4*). This correlation structure contrasts the aperiodic, network bursting activity that is a common feature of developing neural circuits in vivo and in vitro (*O'Donovan et al., 1998*; *Feller, 1999*; *Wagenaar et al., 2006*). We found that both excitatory and inhibitory on-off control were better able to preserve spontaneous activity correlations than PI control (*Figure 5E*; *Figure 5—figure supplement 2C*). However, for targets that required higher excitatory stimulation rates during on-off control, a periodic correlation structure emerged (*Figure 5E*).

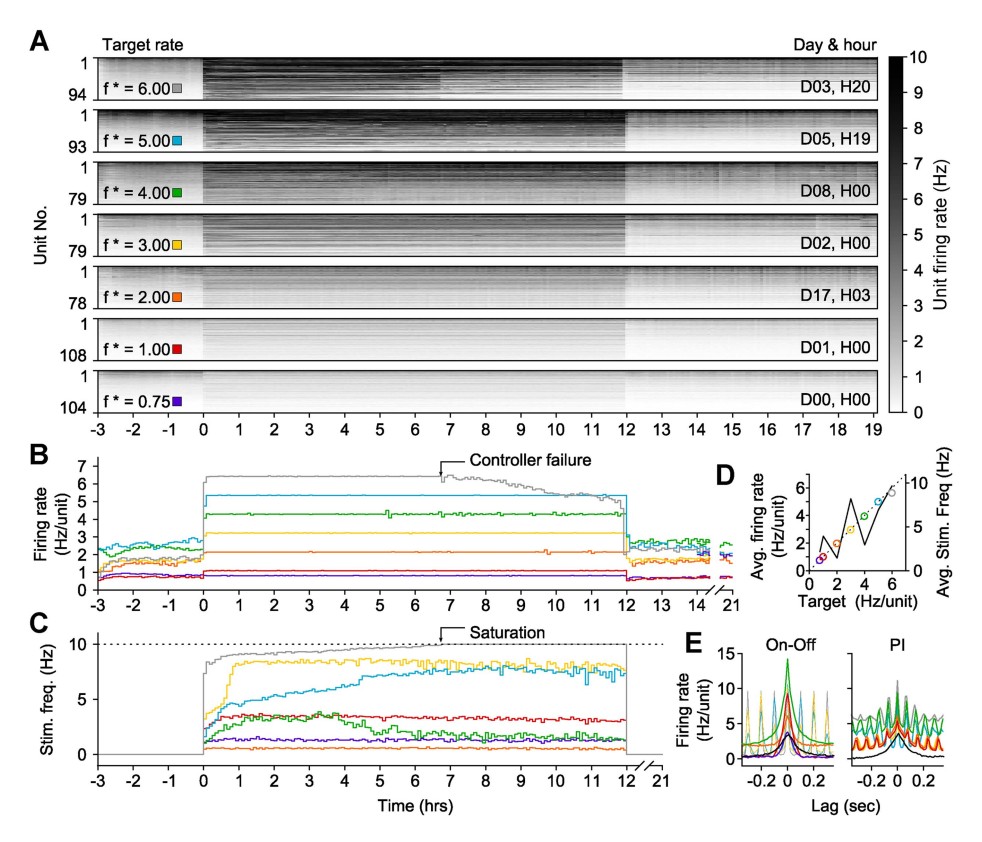

**Figure 5**. On-off feedback control of population firing rate over 12-hr epochs. (All data presented in this figure were obtained from a single culture over the course of ~3 weeks.) (**A**) Firing rates of detected units during 12-hr control periods are represented using the grey-scale to the right. At time 0, the on-off controller was engaged and the average network rate was clamped firing to the target rate indicated to the left of each chart. The day and hour of each protocol, relative to the first experiment, is shown to the right. Units are sorted by their mean firing rate during the 3-hr period prior to closed-loop control. (**B**) The network firing rate during each control epoch (5-min bins). The color map corresponds to the target rates shown in (**A**). (**C**) Closed-loop stimulation frequency over the course of the 12-hr clamp. For a target rate of 6 Hz/unit, the controller saturated at the maximal frequency of 10 Hz at around 7 hr into the control epoch, and target tracking failed as a result. (**D**) Time- and unit-averaged firing rates (colors, left axis) and control signal (black, right axis) across each 12-hr clamping period. The dotted line is identity. (**E**) The average cross-correlation function between 30 randomly selected units during on-off or PI control are plotted for each target rate. The correlation function for spontaneous activity is shown in black. When low stimulation frequencies were required, the unimodal correlation structure of spontaneous activity was preserved using on-off control.

The following figure supplements are available for figure 5:

**Figure supplement 1**. Characteristics of on-off control over weeks in vitro.

**Figure supplement 2**. In vitro inhibitory on-off control using Arch3.0.

## Using on-off control to reveal the direct role of neurotransmission in regulation of network excitability

Previous in vitro studies have shown that chronic elevation in network activity using GABAergic transmission blockers leads to a homeostatic reduction in firing rate following relief from blockade (*Turrigiano et al., 1998*). Conversely, chronic reductions in activity using glutamatergic blockers elicit homeostatic increases in firing rate (*Corner et al., 2002*). Interestingly, we did not observe homeostatic changes in firing rate following prolonged increases or decreases in network spiking

activity during on-off feedback control (*Figure 5*; *Figure 5—figure supplement 1C* and *Figure 5—figure supplement 2B*). This suggests that altered synaptic transmission and altered spiking activity may have distinct effects on homeostatic regulation of network excitability.

To test this possibility, we used on-off control to decouple the effects of prolonged glutamatergic or GABAergic blockade from changes in firing rate. We treated networks with CNQX (2 cultures), AP5 (2 cultures), or bicuculline (1 culture), and then used on-off control to restore pre-drug firing rates during 24-hr (CNQX or AP5) or 3-hr (bicuculline) periods. In all cases, the controller effectively clamped firing rates to pre-drug levels (*Figure 6A.v,B.v,C.v*). 10 min of activity were recorded in the presence of each drug just before and after each clamp period. As expected, application of CNQX or AP5 caused marked reductions in network spiking activity compared to the pre-drug levels and these reduced activity levels were sustained upon relief from on-off control (*Figure 6A.v,B.v*). Meanwhile, bicuculline greatly increased firing rate, both before and after the clamp period (*Figure 6C.v*).

The mean stimulation frequency remained low during excitatory on-off control in the presence of CNQX or AP5 (*Figure 6A.iv,B.iv*); CNQX: 0.72 and 0.19 Hz, AP5: 0.21 and 0.71 Hz. For CNQX, pre-drug network activity correlations and burst shape were largely maintained during on-off control (*Figure 6A.vi–vii*). For AP5, the unit-to-unit correlation time and burst duration were shorter than pre-drug levels (*Figure 6B.vi–vii*). This is likely due to the prominent role of NMDA receptors in signal propagation in dissociated cortical networks (*Nakanishi and Kukita, 1998*). Following drug removal, spontaneous firing was elevated compared to pre-drug levels (CNQX: +86.4 and +21.1%, AP5: +157.4 and +273.3%).

Inhibitory on-off control using Arch3.0 was used to clamp firing to pre-drug levels during bicuculline treatment ('Materials and methods'). Bicuculline greatly increased network firing correlations and burst duration compared to spontaneous activity levels. During the clamp epoch, 'on' to 'off' control transitions reliably triggered large rebound bursts. A rapid closed-loop response truncated rebound bursts via reactivation of Arch-3.0 (*Figure 6C.vi–vii*). After washing bicuculline, firing was reduced by 47.2% compared to pre-drug levels.

Notably, homeostatic changes in spiking levels were observed following prolonged glutamatergic or GABAergic blockade even though network firing rates were maintained at pre-drug levels during the treatment windows. This indicates that changes in firing rate were not required to drive compensatory alterations in network excitability. Instead, homeostatic alterations of network excitability were triggered directly by suppressed synaptic activity. In line with this result, on-off control has recently been used to show that upward synaptic scaling, the most widely studied form of homeostatic plasticity, is directly induced via reductions in AMPA receptor activation (*Fong et al., 2015*).

## Control of single unit activity during ongoing sensory perturbations, in vivo

We next evaluated the functionality of optogenetic feedback control in the intact rodent brain. The rat vibrissal pathway is a widely studied model of sensory information transduction due to its well defined discrete feed-forward anatomy. Recent findings have revealed the importance of network activity state for gating sensory information in thalamic networks (*Halassa et al., 2014*). We used optogenetic feedback to control background firing state in single units of the ventral posteromedial nucleus (VPm) in anesthetized rats during ongoing vibrissa stimulation. Extracellular recordings of ChR2-expressing thalamocortical units (TCUs) were used to update an integral controller (*Equation 15*) (parameters: $T_i$ = 1 s, $\tau$ = 0.8 s), which determined the continuous intensity of 470 nm light delivered to VPm using an optical fiber (*Figure 7A*; 'Materials and methods').

We used optogenetic feedback to clamp firing rates in TCUs at increasing target levels for 30-s epochs (*Figure 7B*). For > 75% of TCUs, the controller effectively clamped firing over a range of target rates, which varied across cells (e.g., 4–40 Hz vs 18–22 Hz; *Figure 7C*). As in our in vitro PI experiments, the RMS tracking error trended upward with increasing target rates (*Figure 7D*). Because firing rate estimates were derived from single cells instead of population activity, the RMS tracking error was larger than for in vitro control (mean ± SD, 2.0 ± 0.8 Hz). Controller settling time was not correlated to the target firing rate and was shorter than for in vitro control (mean ± SD, 3.3 ± 3.2 s).

The optical power required for successful control varied greatly from cell to cell (*Figure 7E*). For instance, the mean optical intensity required for successful control at 16 Hz, the most widely achieved target in our sample, spanned nearly two orders of magnitude (0.66–22.92 mW mm$^{-2}$; *Figure 7E*).

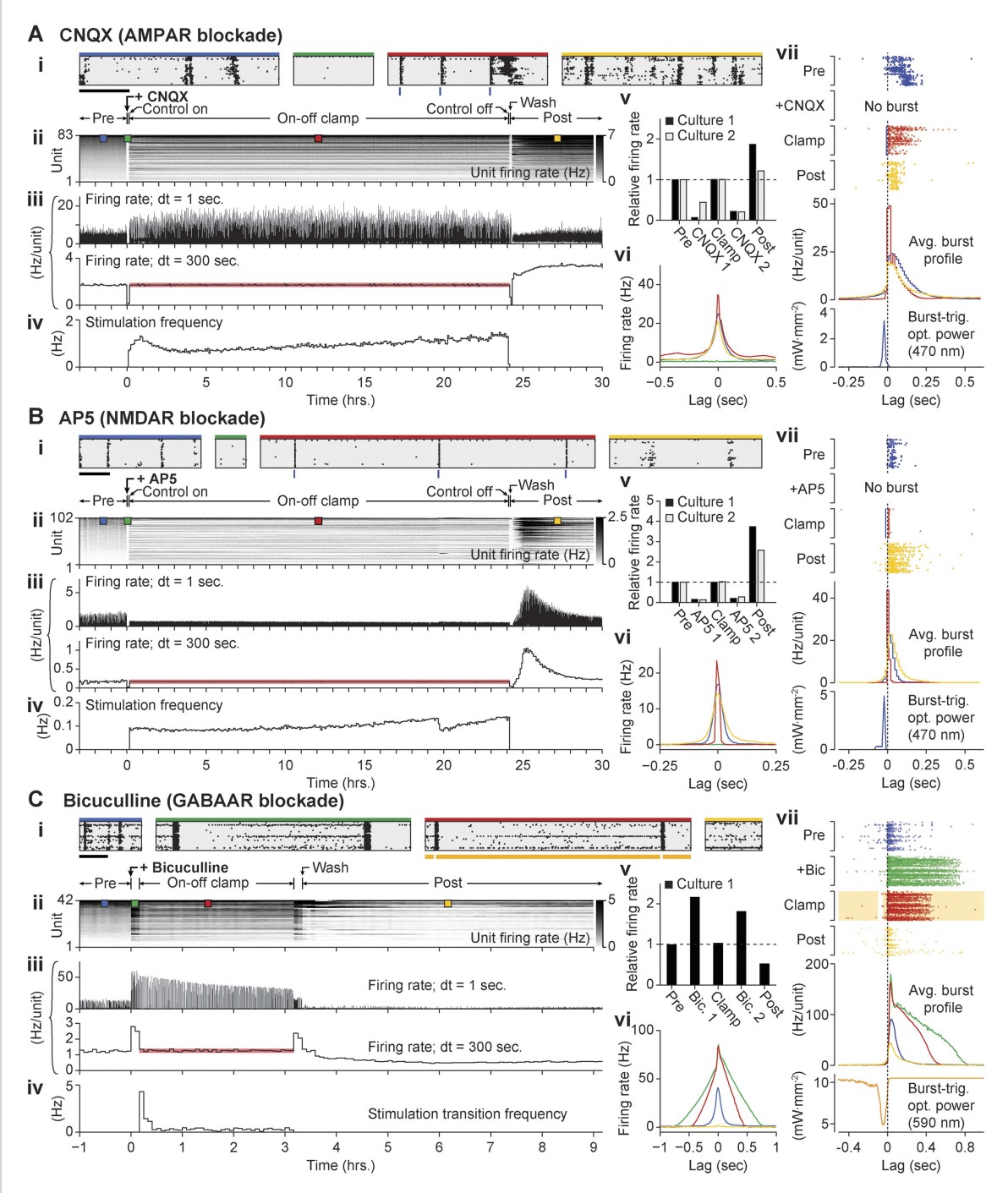

**Figure 6**. Decoupling spiking and neurotransmission using on-off feedback control. (**A**) Summary of a 24-hr AMPAergic neurotransmission/spiking decoupling protocol. (**A.i**) Rastergrams show zoomed portions of spiking activity taken from discrete times during the experiment. Top color bars indicates recording epoch. Blue bars beneath indicate stimulus times. Horizontal scale bar, 1 s (**A.ii**) Firing rate histogram for the duration of the 33-hr recording for each unit, using 5-min bins. Firing levels are indicated by the grey-scale to the right. CNQX (AMPAergic receptor antagonist) was added at time 0 and removed 24 hr and 10 min later. Closed-loop stimulation began 5 min after CNQX addition and lasted 24 hr. Colored boxes indicate the location of the data used in the zoomed rastergrams, crosscorrelograms, and burst profiles. (**A.iii**) The average unit firing rate using 1-s bins and 5-min bins. The red line indicates the target rate. (**A.iv**) Closed-loop stimulation frequency. (**A.v**) Time- and unit-averaged firing rates for each epoch, normalized to the pre-drug firing level. The 'post' firing rate was evaluated over 6 hr following the drug wash. (**A.vi**) The average unit to unit crosscorrelogram for each control epoch. (**A.vii**) Example burst ratergrams, average burst profiles, and burst-triggered stimulus optical intensity for each control epoch. The location of the data used to calculate (**A.vi**) and (**A.vii**) is indicated by the matching colored boxes in (**A.ii**). (**B**, **C**) Same as (**A**) but using AP5 (**B**) or bicuculline (**C**) to

*Figure 6. Continued*

block NMDAergic and GABAergic neurotransmission, respectively. For bicuculline, the firing rate was clamped over a 3 hr period. The changes in spontaneous firing levels before on-off control for each culture were: CNQX, −93.8 and −56.3%; AP5, −87.3 and −84.5%; bicuculline, +116.2%, and upon relief from on-off control: CNQX, −78.7 and −80.0%; AP5, −73.0 and −80.4%; bicuculline: +81.3%.

This suggests that open-loop optical stimuli would not result in consistent firing rates. Indeed, we found that linear increases in optical intensity resulted in extremely variable and temporally non-stationary firing over trials and units (*Figure 7—figure supplement 1*). Further, we examined whether successful control could be achieved by locking the stimulator at a static optical power once the controller stabilized. Halfway through each 30-s trial we locked the light signal either at the last output taken by the controller or the average control signal during first half of the trial (6 TCUs; *Figure 7—figure supplement 2*). Locking optical power at the mid-trial or trial-averaged level significantly increased RMS tracking error (+204 ± 227% and +238 ± 145%, respectively, p < 10⁻¹⁴ for both). This indicates that continuous stimulus updates are required to exert precise control over neural activity in vivo.

Next we examined how changes to the firing rate filter time-constant, $\tau$, or the integral time-constant, $T_i$ affected control performance (*Figure 7—figure supplement 3*). Increasing $\tau$ (from 0.8 to 1.6 s) introduced lag into the control loop, causing overshoot, decreased stability, and a significant increase in RMS tracking error (+57%, p = 1.4 × 10⁻⁶). Conversely, lowering $\tau$ (from 0.8 to 0.16 s) decreased overshoot and significantly reduced RMS tracking error (−46.9%, p = 1.2 × 10⁻¹⁰). Increasing $T_i$ (from 1 to 10 s) did not significantly affect RMS tracking error (p = 0.088), but caused over-damping and increased controller settling time. Reducing $T_i$ (from 1 to 0.1 s) significantly decreased RMS tracking error (−18%, p = 0.011) and introduced overshoot during control onset. Together, these results indicate that low-latency feedback and a short integral time constant improve the performance of firing rate control in vivo. In this experiment, $\tau$ = 0.16 s and $T_i$ = 1 s gave the best performance in terms of RMS error.

In the awake animal, sensory thalamic spike trains tend to exhibit irregular firing with interval statistics close to those of a Poisson process (*Poggio and Viernstein, 1964*). Drugs used for anesthesia have profound effects on evoked and background firing (*Simons et al., 1992*), receptive field properties (*Friedberg et al., 1999*), and subthreshold voltage statistics (*Constantinople and Bruno, 2011*) in the vibrissal pathway. We calculated the coefficient of variation of the interspike interval (ISI) distribution ($CV_{ISI}$) for each TCU. Across target rates and units, we found a $CV_{ISI}$ close to 1, indicating a Poisson-like spiking process (mean ± SD, 1.31 ± 0.33 across targets and units; *Figure 7F*). In comparison, the $CV_{ISI}$ of spontaneous unit activity was significantly elevated (*Figure 7F*, inset) mean ± SD, 1.72 ± 0.43; p = 0.043), likely due to an increased propensity for burst firing in sensory thalamus during anesthesia and non-alert states (*Stoelzel et al., 2009*). This suggests that optogenetic feedback control using continuously modulated input can be used to mimic alert, Poisson-like spiking statistics in anesthetized animals.

Finally, we tested whether the controller could clamp firing during ongoing sensory drive. We recorded from TCUs that were responsive to punctate deflections of the corresponding primary vibrissa (5 TCUs, 8° at ~1600 deg. s⁻¹, 10 Hz; *Figure 8—figure supplement 1*; 'Materials and methods'). Vibrissa deflections evoked stimulus-locked spike trains both in the presence and absence of closed-loop control (*Figure 8*, *Figure 8—figure supplement 1*). However, whisker stimuli resulted in little or no performance degradation of firing rate control in terms of mean TCU firing rate (*Figure 8B*; p = 0.09) or RMS tracking error (*Figure 8C*; p = 0.73) compared to control without whisker input. To maintain control during sensory stimulation, the controller automatically decreased LED stimulus intensity to accommodate sensory drive (*Figure 8D*; mean ± SD, 8.4 ± 5.34 to 4.9 ± 4.8 mW mm⁻², p = 0.024).

Precise temporal spiking patterns carry information in the vibrissal sensory pathway (*Curtis and Kleinfeld, 2009*; *Wang et al., 2010*; *Bruno, 2011*). To characterize the timescales over which the optical controller and whisker stimuli affected firing, we calculated the spike-triggered average (STA) of optical power and whisker position (*Figure 8E,F*). During concurrent whisker stimulation and closed-loop control, the full-width at half-maximum (FWHM) of the STA for optical power and whisker position differed by more than an order of magnitude, indicating that sensory input and optical

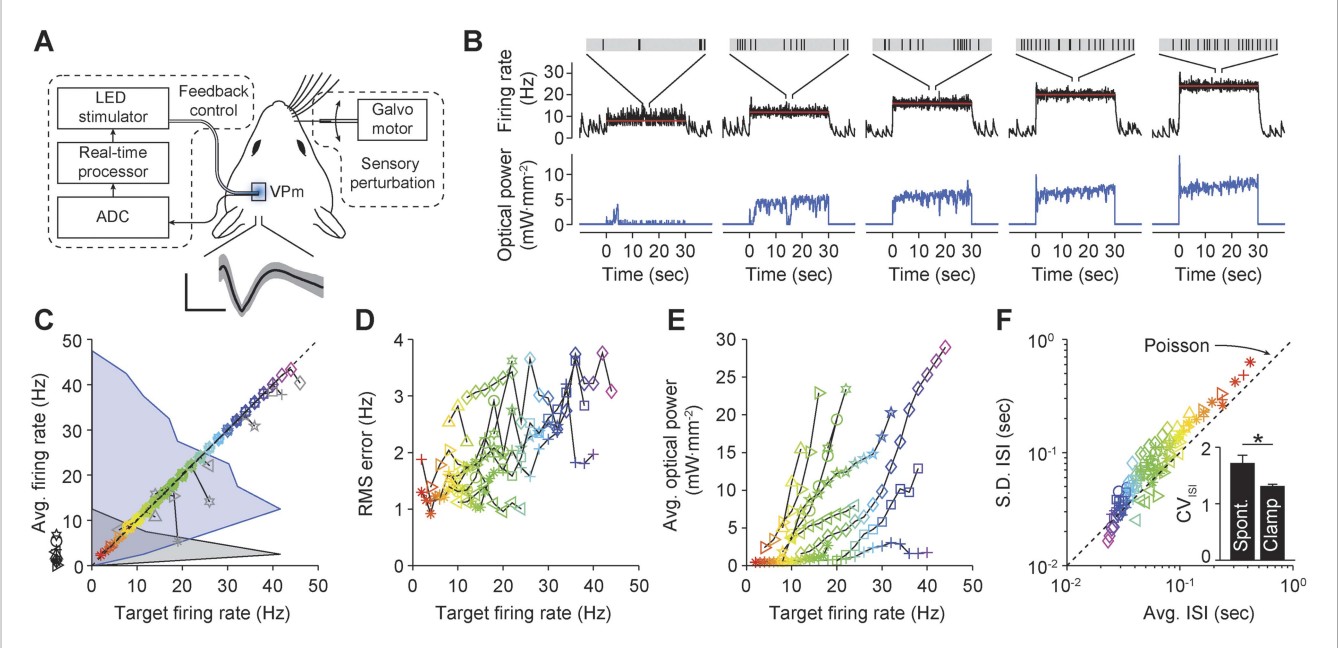

**Figure 7**. Firing rate control of isolated thalamic units, in vivo. (**A**) Single unit extracellular recordings were performed in thalamic VPm and used to update the optical power of the LED stimulator. The primary vibrissa could be deflected along the rostral–caudal plane using a galvanometer-based scanning motor to provide sensory perturbations during closed-loop control. A representative TCU waveform is shown (black line is the mean and the shaded region is ± 1 SD). Vertical and horizontal scale bars represent 100 μV and 1 ms, respectively. (**B**) Single-trial closed-loop firing rate control in the absence of sensory input. Traces show the target firing rate (red), measured firing rate (black), and light power (blue). Inset spike trains display 1 s of activity for each target rate. (**C**) Time-averaged firing rates vs target rates for 10 TCUs (symbols). Data points are color coded according to the target rate. Black symbols at left indicate spontaneous firing levels prior of closed-loop control. Grey symbols indicate control failure. Data points derived from a single TCU are connected with a line. Shaded areas are peak-normalized histograms of spontaneous firing rates (black) and successfully controlled firing rates (blue) across units. (**D**) RMS tracking error for each target rate. (**E**) Average light power required for each target rate. (**F**) Mean vs standard deviation of the ISI distribution for each target rate. The dotted identity line indicates Poisson firing statistics. Inset bar chart shows the mean $CV_{ISI}$ across units during spontaneous and clamped firing. *p = 0.043; $t$-test.

The following figure supplements are available for figure 7:

**Figure supplement 1**. Open-loop application of precisely defined optical stimuli results in highly variable, non-stationary evoked firing levels in the intact rat vibrissa system.

**Figure supplement 2**. Continuous real-time update of optical intensity is required for accurate firing rate control in the intact rat vibrissa system.

**Figure supplement 3**. Effects of firing-rate filter time-constant ($\tau$) and integral time-constant ($T_i$) on performance of integral control in vivo.

control signals affected firing on distinct timescales (*Figure 8G*; mean ± SD FWHM of STA: optical power, 0.46 ± 0.2 s vs whisker position, 18.3 ± 5.5 ms, p = 0.0079). Therefore, optogenetic feedback control using continuously modulated input provides a means to control baseline firing state without distorting the fine-scale temporal structure of sensory-evoked spike trains. Punctate vibrissa stimuli rearrange spike times instead of introducing additional spikes, allowing the temporal correlations in the firing rate to dictate the downstream response rather than the magnitude of the firing rate. This effect is reminiscent of firing modulation by free air whisking in vibrissa primary sensory cortex (*Curtis and Kleinfeld, 2009*) and contrasts the effect of pulsed optogenetic stimulation, which transiently overrides the influence of sensory input on the temporal structure of spike trains.

## Discussion

All previous methods involving closed-loop or activity-guided optogenetic stimulation have used a physiological measurement to trigger static optical stimuli without corrective action or modification of stimulus parameters following stimulus onset (*Leifer et al., 2011*; *Stirman et al., 2011*; *Paz et al., 2012*;

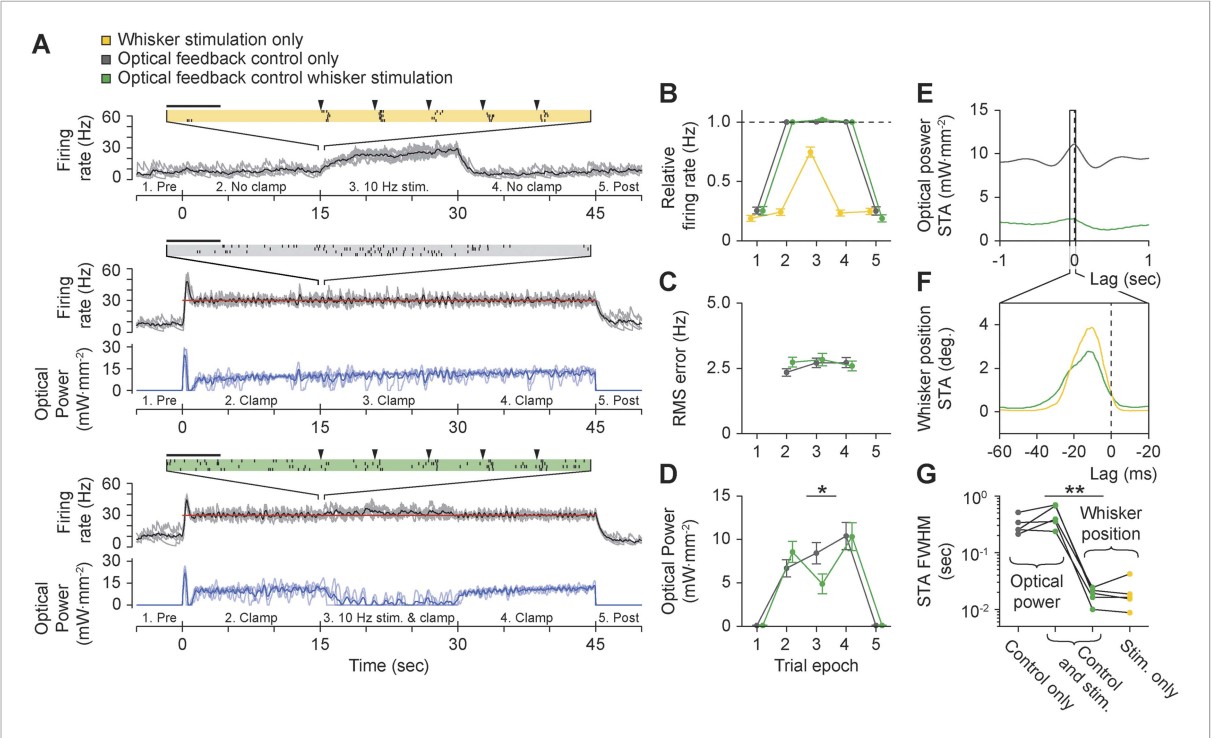

**Figure 8**. Using optogenetic feedback to control thalamic activity state during ongoing sensory input. (**A**) Real-time control of thalamic firing levels during external sensory drive. The firing rate of a single TCU cell (grey lines: single trials; black lines, average) is shown for three interleaved protocol types: 15-s trains of whisker stimuli (yellow), 45-s closed-loop control periods in the absence of whisker input (grey), and closed-loop control during concurrent whisker stimulation (green). (*Top*) 10 Hz whisker deflections occurred from 15–30 s within each trial (black triangles). Inset raster plot shows spikes times for 4 trials at the onset of whisker stimulation. (*Middle*) TCU firing was clamped at 30 Hz (red line) for the duration of each trial. Blue lines show the optical control signal (light blue: single trials; dark blue: average). (*Bottom*) TCU firing was clamped at 30 Hz (red line) for the duration of each trial and whisker stimuli were delivered from 15–30 s within each trial. Horizontal scale bars on the firing rasters indicate 100 ms. (**B**) mean relative (measured/target) firing rates, (**C**) mean RMS tracking errors, and (**D**) mean optical power across trials and units. Values are shown for each of the 5 trial epochs indicated on the abscissa axis of each time series in (**A**). Error bars indicate ± SEM. (**E**) Spike-triggered average (STA) optical power and (**F**) STA whisker position for the TCU shown in (**A**). Note the difference in time scales between (**E**) and (**F**). (**G**) FWHM of the STA for each TCU across trial types. Sample sizes: 5 TCUs, 3 to 5 applications of each protocol type per unit. *p = 0.024, **p = 0.0079; Mann–Whitney U Test.

The following figure supplement is available for figure 8:

**Figure supplement 1**. Spike waveforms and autocorrelograms of TCUs used for concurrent optogenetic feedback control and whisker stimulation in intact rats.

*Krook-Magnuson et al., 2013*; *O'Connor et al., 2013*; *Siegle and Wilson, 2014*). In contrast, the optoclamp continuously updates stimulus intensity and frequency in real-time to enable precise control of neural firing in cultured networks and single cells in vivo and therefore provides a foundation for techniques aiming to achieved true closed-loop optical control of neural activity (*Grosenick et al., 2015*). We have shown that this form of optogenetic feedback control is capable of clamping network firing rates using different stimulation protocols (*Figure 4*) and control algorithms (*Equations 4, 12, 13, 14*) over a wide range of controller parameters (*Figure 2—figure supplement 3* through *Figure 2—figure supplement 5*, *Figure 7—figure supplement 3*) and timescales (*Figure 2* vs *Figure 5* through *Figure 6*). The optoclamp's ability to control firing levels even during various synaptic (*Figure 6*; *Figure 2—figure supplement 6*) and sensory (*Figure 8*) perturbations demonstrates its robustness and provides evidence of the method's suitability in a range of experiments. Finally, we found that even when using the same controller and target firing rate, successful control often required vastly different intensities of optical stimulation across experiments and experimental preparations (*Figure 9*). This suggests that closed-loop regulation of optical stimulation intensity may be a requirment, rather than a benefit, for evoking consistent activity levels across experiments.

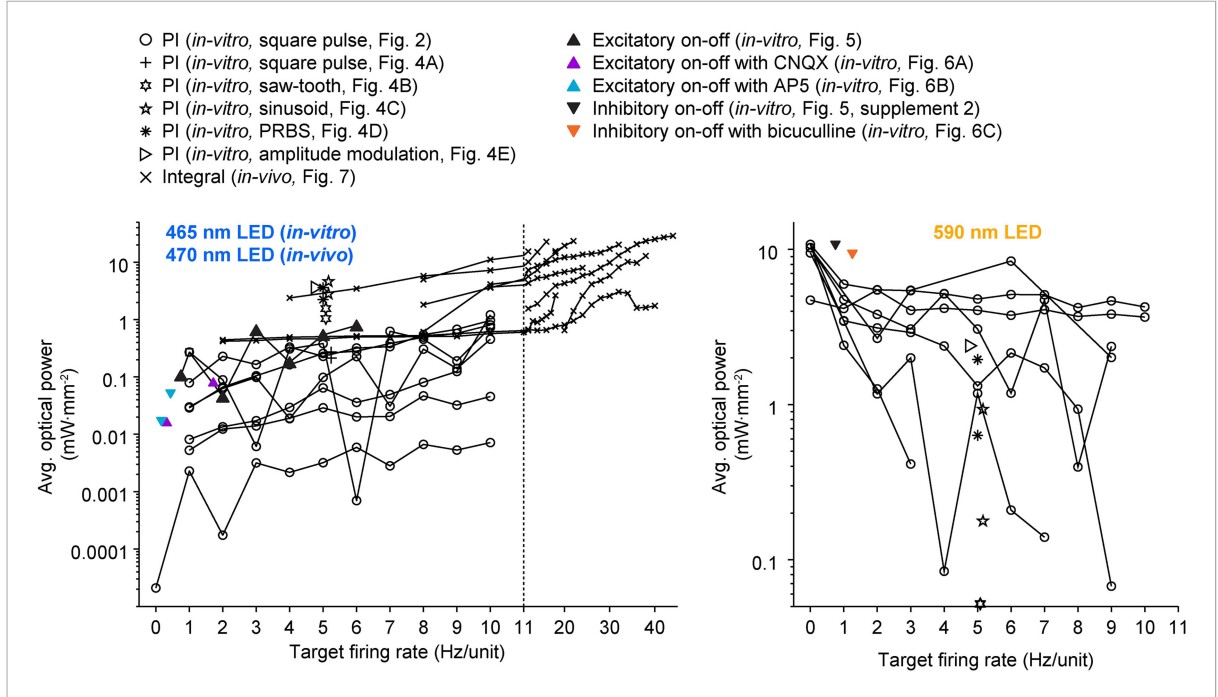

**Figure 9**. A wide range of optical power was required during successful closed-loop control in vitro and in vivo. The time-averaged power density of blue (left plot) and yellow (right plot) light vs the corresponding target firing rate is shown for all control algorithms and experimental preparations used in the paper. Lines connect data points derived from the same culture (in-vitro data) or unit (in-vivo data; note the log scales). Only successful control trials are shown. The light intensity required during closed-loop control varied across orders of magnitude and depended on the target rate, control algorithm, stimulus waveform, type of neural preparation being controlled, and variability in cell-to-cell and culture-to-culture excitability. This highlights the ability of closed-loop control to compensate for the experimental variability across preparations, equipment, and algorithms, as well as the intrinsic variability in neural circuits, to achieve a target activity level.

Like the optoclamp, the voltage clamp relies on a continuously updated feedback loop to control neural activity. Although the scale of neural activity controlled by these two methods differs greatly (population firing rate vs membrane potential), comparing the optoclamp to the voltage clamp is useful for illustrating the power, potential uses, and shortcomings of optogenetic feedback control in its current form, and to highlight how the technique might be extended to allow control over features of neural activity other than firing levels.

Two capabilities of the voltage clamp technique have made it a ubiquitous tool for characterizing neuronal excitability and synaptic dynamics. First, by continuously servoing the current injected into a cell in order to maintain a target membrane potential, the injected current mirrors underlying synaptic or intrinsic conductances that are difficult to measure directly. Second, because the voltage clamp can precisely control the membrane potential, it can decouple strong causal relationships that exist between the membrane voltage and other variables. For example, measuring the open-loop relationship between the membrane voltage and voltage-dependent conductances is impossible in the unclamped cell since these variables feed back onto one another. The voltage clamp breaks this circular dependence in order to systematically examine the strength of conductances at fixed voltage levels.

The optoclamp affords analogous capabilities for examining network dynamics. For instance, population-level excitability in an unclamped network must be inferred through measures of firing activity, which in turn affects network excitability. By continuously updating the intensity of optical stimulation to clamp network firing at set levels, information about network excitability and afferent input can be read directly from optical control signals (*Figure 3C*). One shortcoming of this feature is that it is difficult to identify the origin of changes in network excitability that appear in optical control signals since they result from the aggregate effect of many different processes. However, parallel limitations also exist for the voltage clamp. For example, when using voltage clamp, it can be difficult

to distinguish contributions from different neurotransmitter systems and intrinsic conductances or to deduce the relationship between synaptic currents measured at the recording site vs at the site of receptor binding. Another issue with both optical clamping and the voltage clamp is that the control signal can be contaminated by non-ideal aspects of the recording or stimulation setup. For instance, opsin desensitization could affect the intensity of light required for the optoclamp to maintain firing levels. Likewise, space clamp and seal stability issues that arise during voltage clamp influence the amount of current required to hold a target membrane potential.

In the case of the voltage clamp, these issues are partially compensated for by using secondary measurements of seal quality and by employing the voltage clamp in concert with drugs or genetic manipulations that help to isolate particular synaptic and intrinsic conductances. Different neural circuits have distinct population-level dynamics, connectivity, neurotransmitter systems, cellular constituents, and opsin expression properties. Therefore, analogous to the voltage clamp, the optoclamp should be used in combination with auxiliary techniques that can compensate for imperfections in the control system and allow the mechanistic underpinnings of changes in excitability that occur during clamping to be isolated. For instance, the effects of opsin desensitization and bandwidth can be factored out of optical control signals such that the control signals accurately reflect changes in network excitability during clamping (*Tchumatchenko et al., 2013*). Further, the optoclamp can be combined with existing pharmacological, genetic, or electrophysiological tools to isolate specific mechanisms that influence network excitability following clamping (*Fong et al., 2015*).

In addition to providing a means to quantify network excitability, like the voltage clamp, optogenetic feedback control is capable of decoupling causally related variables of circuit activation. Using two case studies, operating on very different time-scales, we have demonstrated that the optoclamp is capable of decoupling thalamocortical cell baseline state from fine-scale sensory-evoked firing patterns in vivo (*Figure 8*) and decoupling network firing levels from various forms of neurotransmission (*Figure 6*). The former is especially relevant given recent focus on the state-dependent nature of thalamic coding (*Halassa et al., 2014*), and the need for stimulation technologies capable of controlling non-stationary neural dynamics in order to inject meaningful sensory information into damaged circuits (*Stanley, 2013*). Decoupling firing levels from variables to which they are normally causally intertwined can also help clarify causal relationships between firing and other factors that influence network excitability. For instance, determining the independent roles of neurotransmission and spiking on various forms of plasticity has proven challenging since manipulation of neurotransmission invariably affects spiking levels, and vice versa. To overcome this, Fong et al. recently used the optoclamp to decouple network firing levels from long-term AMPAergic neurotransmission blockade to show that upward synaptic scaling is directly triggered by reduced AMPAergic transmission without relying on changes in spiking activity (*Fong et al., 2015*). This provides a plasticity mechanism to explain the increases in network firing levels we witnessed following long-term CNQX treatment even when firing was clamped to pre-drug levels for the duration of drug application (*Figure 6A*).

The control algorithms and technologies presented here are simple, straightforward, and widely available. Our implementations of the optoclamp are quite reliable for controlling neural activity in cultured networks and thalamic cells in vivo and the technique's simplicity lowers the barrier for its adoption. However, there are several avenues for further development of the method we have presented. For instance, we used a pre-sorting procedure to identify neurons used for real-time firing rate estimation in subsequent clamping periods. During periods of optical stimulation, neurons that were silent during the pre-sorting routine may be activated. If these cells had systematically different firing characteristics the sorted units used for firing rate estimation, then our procedure would lead to a biased estimate of network firing levels. Although we found no evidence that this was the case in our networks, the situation might differ for other neural preparations or brain regions. If this issue were to arise, spike sorting could be removed entirely and firing could be normalized by the number of recording channels, provided that the channel count is sufficiently high.

Further, the incorporation of more sophisticated activity measurement techniques, stimulation technologies, and control algorithms will enable improvements in control performance and broaden the applicability of optogenetic feedback control to different experimental contexts (*Grosenick et al., 2015*). For example, the incorporation of spatial light modulation would allow optical inputs to be steered towards the spike initiation zones of individual cells in order to minimize light exposure (*Figure 9*) and abnormal conductances, and potentially enable complex

system identification alogrithms to be introduced into the feedback loop (*Grosenick et al., 2015*). Additionally, decreased feedback latency or the addition of predictive elements to the feedback loop may enable control over rapid sensory or motor events. There are two options to obtain accurate firing rate measures over very short timescales (e.g., that of individual whisker perturbations). The first option is to sample a very large population of neurons such that small time bins will have an adequate number of spikes for accurate estimation of population firing levels. This large population measure would need to be combined with a sub-millisecond feedback loop and opsins with extremely fast kinetics. Alternatively, if an accurate model of feedforward network dynamics could be incorporated into the controller, reliable control over fast events might be possible with a modestly sized population of cells, a slower feedback loop, and standard opsin variants. This is a viable approach in circuits for which predictive models of feedforward network dynamics are available, such as early visual, auditory, and vibrissal pathways (*Wu et al., 2006*; *Millard et al., 2013*), or, for which accurate input/output relationships can be deduced in situ using real-time system identification (*Grosenick et al., 2015*). Additionally, control algorithms that incorporate models of feed-forward neural dynamics will be more capable of stabilizing firing in unstable circuits, such as epileptic networks, without total cessation of ongoing activity.

We demonstrated that optical waveforms with very different temporal characteristics could be used to successfully control population firing rate in vitro while having markedly different effects on spiking correlation and synchrony across individual units (*Figure 4*). However, in the optoclamp's current form, higher-order temporal characteristics, such as the unit-to-unit firing correlation, are not actively controlled features of neural activity. Therefore, during clamping periods, these features of population firing will be dependent on network architecture, the identity and percentage the cells expressing opsins, and opsin dynamics. Improvements on our technique might treat higher order features of network activity, such as unit-to-unit correlation and synchrony, as secondary control targets. In this case, the controller would use real-time measures of higher order firing statistics to adjust the spatial and temporal characteristics of optical stimuli in order to enforce a particular firing structure, such as the heterogeneity of activity levels across cells and/or temporal variance of firing activity (e.g., regular firing vs bursting). Optogenetic feedback control could also be incorporated into more complex experimental contexts. For example, firing rate control could be made contingent on specific behaviors or complex spatiotemporal activity patterns associated with specific behaviors (*Zhang et al., 1998*), in order to introduce fictive reward or neuromodulatory signals to influence learning or alleviate pathological activity. In particular, optoclamping cortical activity to replace lost neuromodulatory tone is a potentially exciting future avenue for treating Parkinson's disease (*Beuter et al., 2014*).

Recently, several 'all-optical' electrophysiology techniques have been introduced to simultaneously measure neural activity via genetically encoded calcium sensors and optogenetically inject currents at single cell resolution (*Packer et al., 2014*; *Rickgauer et al., 2014*). If combined with real-time control, these techniques could offer the ability to optically clamp activity levels in specified subnetworks with far greater specificity than is afforded using electrodes. Perhaps most exciting, recent improvements in microbial rhodopsins for simultaneous voltage indication and optogenetic stimulation provide a means for all-optical measurement and perturbation of the membrane voltage at subcellular resolution and millisecond timescales (*Flytzanis et al., 2014*; *Hochbaum et al., 2014*). These tools even allow simultaneous sensing and actuation using a single opsin. Using opsins to both measure and actuate voltage within a feedback control loop will open the possibility of voltage clamping arbitrary populations of cells without puncturing their cell bodies. This would enable an unprecedented improvement of fine-scale control and measurement of neural circuit activation in vitro and, with specialized optics, in vivo.

In summary, we have performed a systematic and extensive investigation of how optogenetic feedback control can be used to precisely control neuronal firing levels during perturbations that strongly affect network excitability, across time scales ranging from seconds to days, both in vitro and in vivo. The functionality of our technique across control parameters, algorithms, preparations, and firing rate measures (network vs single units) indicates the robustness and general applicability of the technique to different experimental contexts. When combined with secondary genetic, pharmacological, or behavioral manipulations and tailored to particular experimental contexts using suitable control algorithms, we envision the use of optogenetic feedback control in a multitude of experimental and clinical contexts requiring precise control of neuronal activity. For these reasons, we believe that the optoclamp is a powerful addition to the expanding optogenetic toolbox and will

improve and accelerate the study of neural control of motor action, sensory encoding and adaptation, neuromodulation, and activity homeostasis.

## Materials and methods

### Statistics

All statistical analyses were performed using MATLAB (MathWorks, Natick, MA). For tests between two groups, we first used a Lilliefors test ($\alpha = 0.05$) to determine if sample distributions were normally distributed. If the null hypothesis of normality was rejected for one or both sample distribution(s), we performed a Wilcoxon signed-rank test ($\alpha = 0.05$). Otherwise we used a paired t-test ($\alpha = 0.05$). We used paired tests because our samples were 'matched' (i.e., the same culture or cell was examined in two different experimental conditions).

For tests involving multiple comparisons across three or more groups, we first used a Lilliefors test ($\alpha = 0.05$) to determine if the sample distributions were normally distributed. If the null hypothesis of normality was rejected for one or more sample distribution(s), we performed a Kruskal–Wallis one-way analysis of variance. Otherwise, we performed standard one-way ANOVA. Post-hoc hypothesis testing was performed using the Bonferroni correction to control the familywise error rate in order to determine which pairs were significantly different. We used t-tests if sample distributions were Gaussian and Mann–Whitney U tests otherwise. Adjusted p-values are reported in figure captions and the text.

### In vitro procedures

#### Cell culture

Whole neocortex was isolated from embryonic day 18 (E18) rats in accordance with the National Research Council's Guide for the care and use of laboratory animals using a protocol approved by the Georgia Tech IACUC. Cortical tissue was digested in 20 U ml$^{-1}$ papain (Sigma-Aldrich, St. Louis, MO) diluted in a culturing medium described in (*Jimbo et al., 1999*), but without antibiotics or antimycotics. Following enzymatic digestion, cells were dissociated mechanically using 3 to 5 passes through a 1 ml conical pipette tip, and diluted to 2500 cells μl$^{-1}$. MEAs with 200 μm electrode spacing, 30 μm electrode diameter (Multichannel Systems, Reutlingen, Germany) were sterilized using 70% ethanol and exposure to UV light, and precoated with laminin. Approximately 50,000 cells in a 20 μl drop were plated onto a ~2 mm diameter area over the array, resulting in ~2500 cells mm$^{-2}$ on the culturing surface. The culturing well of each MEA was sealed with a fluorinated ethylene-propylene membrane (*Potter and DeMarse, 2001*). Experiments and culture storage were carried out in an incubator regulated to 35°C, 5% $CO_2$, 65% relative humidity. The details of our culturing methods are described in (*Hales et al., 2010*). All experiments were carried out on cultures that were 3–4 weeks old.

#### Viral transfections

Concentrated aliquots of AAV2-CaMKll$\alpha$-hChR2(H134R)-mCherry, AAV2-CaMKll$\alpha$-eNpHR3.0-eYFP, and AAV2-hSyn-eArch3.0-eYFP were produced by the University of Carolina Chapel Hill Vector Core using DNA provided by Karl Deisseroth (Stanford University). When cultures reached 1 to 5 days in vitro (DIV), viral aliquots were diluted to $1 \times 10^{12}$ c.f.u. ml$^{-1}$ and 1 μl was added to 1 ml culturing medium. Infected cultures were incubated for 3 days with the viral solution before the culturing medium was exchanged. To evaluate this protocol, we monitored the fluorescent signal of hChR2(H134R)-mCherry's reporter protein in 3 sister cultures over the days post infection using an LSM510 confocal microscope (Carl Zeiss AG, Oberkochen, Germany). Identical laser power and imaging settings were used for each imaging session. The fluorescent signal increased monotonically before plateauing at ~3 weeks in vitro (*Figure 1—figure supplement 2A,B*). Additionally, the functional reactivity of the cultures to 465 nm and 590 nm optical stimuli was probed in the weeks following infection (*Figure 1—figure supplement 2C*). The ability of ChR2$_R$, eNpHR3.0, and Arch3.0 to affect network firing levels mirrored the expression time course of the marker proteins.

#### Multichannel electrophysiology

Microelectrode voltages were amplified and bandpass filtered between 1 Hz and 5 kHz using a 60 channel MEA60 analog amplifier (Multichannel Systems, Reutlingen, Germany). When stored in a 35°C incubator, the temperature of the amplifier exceeded 37°C. Therefore, the culture was regulated to 35°C using a servo-controlled (Modular One Technology, Parker, TX) custom solid state Peltier cooler

mounted below the recording amplifier (*Figure 1A*). Analog signals were digitized and processed by the NeuroRighter multichannel electrophysiology platform (https://sites.google.com/site/neurorighter/) (*Newman et al., 2013*). Amplified electrode voltages were digitally filtered using a third order Butterworth filter with a passband of 300–5000 Hz. Extracellular action potentials were detected using a voltage threshold of 5 times the RMS noise on each electrode. A spike classifier was trained for each channel by collecting a set of spike waveforms, projecting them into their first two principal components, and fitting a mixture of $K$ Gaussians to the resulting 2D sample distribution using expectation maximization. $K$ was deduced automatically using a minimum description length cost function. Following training, spikes were classified online with a maximal latency of ∼5 ms. The details of NeuroRighter's spike detection/sorting algorithms are presented elsewhere (*Newman et al., 2013*). We note that, given the simplicity of our controllers, they could be easily implemented on any multichannel electrophysiology system capable of rapid, real-time feedback (e.g., open-ephys, http://open-ephys.org/ or Tucker–Davis Technologies bioprocessors, http://www.tdt.com/).

## Optical stimulator

To deliver optical stimuli, we used a custom LED driver (http://www.open-ephys.org/cyclops/) to control a single blue LED (LZ4-00B200, LEDEngin, San Jose, CA) and 3 amber LEDs wired in series (LZ4-00A100, LEDEngin). LEDs were butt-coupled to a 4-to-1 randomized fiber bundle (Schott AG, Mainz, Germany), which then fed light into a custom Köhler illumination train mounted beneath the MEA amplifier (*Figure 1A* and *Figure 1—figure supplement 1*). We confirmed the spatial homogeneity of light at the culturing surface using a BC106-VIS CCD-based beam profiler (Thorlabs, Newton, NJ; *Figure 1—figure supplement 1B*). Because the blue LED was used to deliver complex temporal waveforms (*Figure 4*), we used optical feedback to completely linearize the relationship between the control signal and the LED's radiant intensity (*Tchumatchenko et al., 2013*). Scattered 465 nm light was sampled using a PDA36 amplified photodiode (Thorlabs) fitted with a FB450-40 $450 \pm 22$ nm FWHM optical bandpass filter (Thorlabs) mounted on the Kohler illuminator optical cage (*Figure 1A*). Amplified light power measurements were fed back to the LED driver to linearize the relationship between the control voltage, provided by NeuroRighter, and optical power. Yellow LEDs were driven using a second driver in a constant-current configuration. The static control signal to irradiance relationship for both light sources are shown in *Figure 1—figure supplement 1A,B*. A full design specification for the LED driver is available online (https://github.com/jonnew/cyclops).

## Pharmacology

Each MEA culturing well contained 1.5 ml of culturing medium. To administer synaptic blockers, 100 μl of culturing medium was transferred from the well to a 0.5 ml centrifuge tube and mixed with either 3 μl of 10 mM CNQX (6-cyano-7-nitroquinoxaline-2,3-dione), 3 μl of 25 mM AP5 (amino-5-phosphonovaleric acid), or 3 μl of 10 mM bicuculline. The resulting mixture was returned to the culturing well and pipetted up and down 5 times to arrive at a final concentration of 20 μM CNQX, 50 μM AP5, or 20 μM bicuculline.

## Functional expression

To characterize the ability of $ChR2_R$ to increase population activity, we scanned three parameters of $ChR2_R$ excitation in open-loop: 0.1–5 ms pulse width, 1–40 Hz stimulation frequency, and 0.1–1.5 Amps through a $465 \pm 11$ nm FWHM LED, which corresponds to 1.6–13.4 mW mm$^{-2}$ at the culturing surface in our configuration (*Figure 1—figure supplement 1A*). We found that all three parameters provided smooth, positive, monotonic relationship with the average population firing rate at any point approximately 1 week after viral transduction, and that the functional ability of $ChR2_R$ co-varied with its expression time-course (*Figure 1—figure supplement 2C*). Therefore, we used a single control variable, called $U_C$, to simultaneously modulate the pulse-width, stimulation frequency, and optical power of 465 nm stimulation (*Equations 7–9*).

To characterize the ability of eNpHR3.0 to decrease population firing, we delivered 30 s long stimulus pulses ranging from 0 to 1 Amp to three $590 \pm 10$ nm FWHM LEDs wired in series, throughout development. These LED currents corresponded to ∼1.3–10.8 mW mm$^{-2}$ in our configuration (*Figure 1—figure supplement 1B*). We observed a negative, monotonic relationship between the optical power of the LED and population firing throughout development. Therefore, we defined a control input $U_{eNpHR}$ as the forward diode current of the 590 nm LED (*Equation 10*).

## Feedback controllers

Optogenetic feedback control was implemented using the NeuroRighter plug-in interface, which allows on-the-fly access to NeuroRighter's data streams to user written plug-in code (*Newman et al., 2013*). Every $dt = 4$ ms, the average network firing rate, $f[t]$ was calculated using action potentials produced by sorted units and passed through a first-order averaging filter,

$$f[t] = \alpha r[t] + (1 - \alpha)f[t - dt], \tag{1}$$

where $r[t] =$ no. spikes$/($no. units$\cdot dt)$ is the instantaneous firing rate during the 4 ms bin, averaged across all detected units and the weighting factor,

$$\alpha \approx 1 - \exp(-dt/\tau), \tag{2}$$

is defined using a $\tau = 2.5$ s time constant. The firing rate was then compared to a desired firing rate, $f^*$ (setpoint), and the error between the two,

$$e_f[t] = f^* - f[t], \tag{3}$$

was used to generate stimulus signals using either a PI or on-off control scheme. The PI controller was defined in a recursive form as,

$$u[t] = u[t - 1] + K\left(e_f[t] - e_f[t - 1] + \frac{T_s}{T_i}e_f[t]\right), \tag{4}$$

where $K = 0.1$ is the proportional gain, $T_i = 1$ s is the integral time constant, and $T_s = 0.01$ s is the period of the control loop. $u[t]$ was then transformed into optical stimulus signals according to,

$$U_C = u[t] + \Delta_1, \tag{5}$$

$$U_H = -u[t] + \Delta_2. \tag{6}$$

$\Delta_{1,2} = 0.25$ determine the degree of overlap in $ChR2_R$ and eNpHR3.0 activation, respectively. $U_C$, the control variable for $ChR2_R$, was transformed into pulses of blue light according to,

$$\text{Pulse freq.}_{465\text{ nm}} = 10 \cdot U_C + 10 \text{ (Hz)}, \tag{7}$$

$$\text{Pulse width}_{465\text{ nm}} = 5 \cdot U_C \text{ (ms)}, \tag{8}$$

$$\text{Power}_{465\text{ nm}} = 13.2 \cdot U_C \text{ (mW}\cdot\text{mm}^{-2}). \tag{9}$$

$U_H$, the control variable for eNpHR3.0, was transformed into continuously modulated yellow light according to,

$$\text{LED current}_{590\text{ nm}} = U_H \text{ (Amps)}. \tag{10}$$

$U_C$ and $U_H$ were bounded between 0 and 1 to prevent integral windup and unreasonably high stimulation intensities. 20 s prior to the start of each 60-s control epoch a 10-s train of $U_C = 1.0$ stimuli was applied, which we found increased control stability by preventing oscillations at the start of the control epoch. This conditioning stimulus train is referred to as a 'pre-pulse' in *Figure 3*. If oscillations persisted, we increased $T_i$ from its nominal value of 1 s until the controller stabilized (*Figure 2—figure supplement 4*). The largest value of $T_i$ required for to achieve stable control was 10 s.

The excitatory on-off controller was defined as,

$$I_f[t] = \sum_{k=0}^{t} e_f[k], \tag{11}$$

$$\text{Stim}\left[t\right] = \begin{cases} 5 \text{ ms,} \quad 465 \text{ nm,} \quad 13.2 \text{ mW}\cdot\text{mm}^{-2} \text{ pulse} & \text{if } I_f[t] > 0 \\ \text{Off} \quad \text{Otherwise} \end{cases}, \tag{12}$$

where and $I_f[t]$ is the integrated error signal and a maximal stimulation frequency of 10 Hz was enforced. The inhibitory on-off controller was defined as,

$$\text{Stim}\left[t\right] = \begin{cases} \text{On} & \text{if } I_f[t] < 0 \\ \text{Off} & \text{Otherwise} \end{cases}, \tag{13}$$

where 1.0 A was delivered to the 590 nm LEDs during the 'on' phase, producing ~11.8 mW mm$^{-2}$ at the MEA.

## In vivo procedures

### Experimental preparation

All procedures were approved by the Georgia Institute of Technology Institutional Animal Care and Use Committee and followed guidelines established by the National Institutes of Health. Female sprague-dawley rats (250–300 g) underwent an initial survival surgery, during which the viral vector (AAV2-CaMKllα-hChR2(H134R)-mCherry, UNC Viral Vector Core, Chapel Hill, NC) was delivered to the left thalamus using stereotactic coordinates targeting the ventral posteriomedial nucleus. The viral syringe (Neuros Syringe, Hamilton Laboratory Products, Reno, NV) was slowly lowered to depth (approximately 2 mm/min) where 1 µl of viral vector solution was delivered at 0.2 µl/min. The injection was allowed to sit for 5 min before slowly retracting the syringe to prevent movement of the virus away from the target location. The animals recovered for 3–4 weeks, providing time for $ChR2_R$ expression to reach functional levels.

In a second acute surgery, the rodents were initially anesthetized with 4% isoflurane before transitioning to intravenous administration of fentanyl cocktail (5 µg/kg fentanyl, 150 µg/kg dexmedotodomine, 2 mg/kg midazolam) through the tail vein. Throughout the experiment, measurements of the heart rate, respiratory rate, oxygen saturation, and response to toe pinch stimuli were used to monitor and titrate the depth of anesthesia. Body temperature was maintained at 37°C by a servo-controlled heating blanket (FHC, Bowdoinham, ME). Animals were mounted in a stereotactic frame and a craniotomy was performed over the left parietal cortex to allow access to the ventral postero-medial (VPm) region of the thalamus (coordinates: 2–4 mm posterior to bregma, 2.5–3.5 mm lateral to midline, 4.5–5.5 mm below cortical surface).

### Electrophysiology

The 'optrode' consisted of a multimode optical fiber (105 µm core diameter, 125 µm coating diameter, 0.22 NA, Thorlabs, Newton, NJ) and one tungsten microelectrode (75 µm diameter, FHC). The microelectrodes had an impedance of 1–2 MΩ at 1 kHz. The optical fiber was ground to a fine point, producing a spherical, rather than conical, pattern of light delivery. The optrode was advanced to the ventral posterio-medial (VPm) region of the thalamus using a precision microdrive (Knopf Instruments, Tujunga, CA). Single and multi-unit activity were band-pass filtered between 300 and 5000 Hz and digitized at 24.414 kHz using an RZ2 multichannel bioacquisition system (Tucker Davis Technologies, Alachua, FL). The principal vibrissa was determined by manually deflecting individual whiskers and observing resultant multi-unit activity.

### Whisker stimulation

Whiskers were trimmed at approximately 12 mm from the face, and were inserted into a custom attachment of a feedback-controlled galvo motor (range of motion, ± 20°; bandwidth, 250 Hz; Cambridge Technology) positioned 10 mm from the vibrissa pad. Vibrissae were always deflected in the rostral–caudal plane. Punctate deflections consisted of exponential rising and falling phases (total excursion, 8°; 99% rise and fall times, 5 ms; average angular velocity, 1600° s$^{-1}$).

### Closed-loop optical stimulation

We drove a 470 nm high power LED (LBW5SN, OSRAM Opto Semiconductors GmbH, Regensburg, Germany) using our custom LED driver in optical feedback mode. The LED was butt-coupled to a 125 µM diameter optical fiber (Thorlabs) which was used to deliver light to the VPm thalamus and stimulate the $ChR2_R$-expressing cells. Because we used optical feedback to drive the LED, the control output, $u[t]$, was converted directly into a continuously varying light intensity,

$$Light\ power = G \cdot u[t], \tag{14}$$

where $G$ is a gain parameter that accounts for loss in the coupling from the LED die to the fiber and tuning of the driver circuit. $G$ was measured for each fiber used (*Figure 1—figure supplement 1D*). The controller was implemented on the Tucker–Davis RZ2's digital signal processors using the RPvdsEx graphical programming language. The in-vivo control loop was equivalent to *Equations 1–4* with the exception that $\tau = 0.8$ s and the proportional component was set to zero,

$$u[t] = u[t-1] + \frac{T_s}{T_i} e_f[t] \tag{15}$$

## Acknowledgements

We thank M LaPlaca for providing tissue and JT Shoemaker for performing tissue harvests. JPN, MF, and DCM were each supported by a US National Science Foundation Graduate Research Fellowship. JPN and MF were supported by a US National Science Foundation Integrative Graduate Education and Research Traineeship. CJW is supported by the National Institutes of Health Ruth L Kirschstein National Research Service Award (F31NS089412) and was supported by the National Institute of Health computational neuroscience training grant (1T90DA032466). GBS was supported by US National Science Foundation Collaborative Research in Computational Neuroscience grant IOS-1131948 and US National Institutes of Health National Institute of Neurological Disorders and Stroke grant 2R01NS048285. SMP was supported by US National Science Foundation Emerging Frontiers in Research and Innovation grant 1238097 and US National Institutes of Health National Institute of Neurological Disorders grant 1R01NS079757-01.

## Additional information

### Funding

| Funder | Grant reference | Author |
|---|---|---|
| National Science Foundation (NSF) | Integrative Graduate Education and Research Traineeship | Jonathan P Newman, Ming-fai Fong |
| National Institute of Neurological Disorders and Stroke (NINDS) | Kirschstein National Research Service Award | Clarissa J Whitmire |
| National Science Foundation (NSF) | Collaborative Research in Com- putational Neuroscience grant IOS-1131948 | Garrett B Stanley |
| National Institute of Neurological Disorders and Stroke (NINDS) | 2R01NS048285 | Garrett B Stanley |
| National Institute of Neurological Disorders and Stroke (NINDS) | R01NS085447 | Garrett B Stanley |
| National Science Foundation (NSF) | Emerging Frontiers in Research and Innovation grant 1238097 | Steve M Potter |
| National Institute of Neurological Disorders and Stroke (NINDS) | 1R01NS079757-01 | Steve M Potter |
| National Science Foundation (NSF) | Graduate Research Fellowship | Jonathan P Newman, Ming-fai Fong, Daniel C Millard |
| National Institutes of Health (NIH) | Computational Neuroscience Training Grant (1T90DA032466) | Clarissa J Whitmire |

The funders had no role in study design, data collection and interpretation, or the decision to submit the work for publication.

### Author contributions

JPN, Conception and design, Acquisition of data, Analysis and interpretation of data, Drafting or revising the article; M-F, Conception and design, Acquisition of data, Drafting or revising the article; DCM, CJW, Acquisition of data, Drafting or revising the article; GBS, SMP, Conception and design, Drafting or revising the article

### Author ORCIDs

Jonathan P Newman, http://orcid.org/0000-0002-5425-3340

## Ethics

Animal experimentation: This study was performed in strict accordance with the National Research Council's Guide for the care and use of laboratory animals using protocol A12012 approved by the Georgia Tech IACUC.

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
