## [Decision Letter]

Thank you for sending your work entitled “Optogenetic feedback control of neural activity” for consideration at *eLife*. Your article has been favorably evaluated by Eve Marder (Senior editor) and two reviewers, one of whom is a member of our Board of Reviewing Editors.

The Reviewing editor and the other reviewer discussed their comments before we reached this decision, and the Reviewing editor has assembled the following comments to help you prepare a revised submission.

The overall statements of both reviewers were positive. Indeed, the novelty of the technique for reading out and controlling the activity of neuronal networks, was positively valued. However, both reviewers formulated several main concerns, of which two are most salient.

1) The mechanisms underlying the light-mediated alternation or clamp of neuron excitability are highly dependent on various factors including neuron types expressing ChR2, their cellular and synaptic properties and interconnectivity. It was unclear how these factors may influence the speed and quality of clamping neuronal excitability.

2) One reviewer pointed out that it is unclear how the activity spreads through the network and how it may change the input. Finally, the reviewer pointed out that the speed of the system is a limiting factor in a fast control of network excitability. The question is whether the speed can be improved. In general, a more thorough discussion of the limitations of the techniques should be discussed. Below you will find the major and minor criticism of the two reviewers.

*Reviewer #1*:

Major criticism:

By comparing sinusoidal light application to ChR2 expressing cell populations in comparison to rectangular or triangular light application, the authors observed differences in spike correlations and synchrony in some of the applied light protocols. The problem with this approach is that the resonance behavior of neurons is highly diverse and depends on the intrinsic membrane properties. Thus, any interpretations on the population behavior will depend on the nature of the cells expressing ChR2 and the percent of the contributing neuron types expressing CR2. Thus, the mentioned 'systematic modulation' of neuronal networks at the end of the subsection headed “Proportional-integral control of network firing” is very much dependent on various factors such as cell types, percent of neuron types expressing Chr2, brain area under investigation. This should be discussed in the manuscript.

Minor criticism:

At the end of the first paragraph of the subsection headed “Multi-hour control of firing rates” I would propose to replace the wording 'network plasticity' by 'changes in neuronal network dynamics'.

*Reviewer #2*:

In this manuscript the authors propose a way to measure the input into a neuronal population by quantifying the optogenetic current which is required to keep those cells at a constant firing rate level. They use eNpHR3.0 to balance the lack of inhibition and ChR2 to counteract the lack of excitation in vitro and in vivo. With this method, the authors approximate the input to the neuronal population by the power of the yellow light minus the power of the blue light. Since the study controls for the neuronal firing, the clamp can be used to cancel slow components for several seconds. This was used to keep the neuronal firing rate constant in order to show that the synaptic homeostasis is not dependent on neuronal firing rate. This is a nice application of the method. Overall the paper is well written and the figures are clear. What is missing is a discussion and data about how this method differs from classic patch clamp and how one can deal with the differences. The patch clamp is an important reference since it is relatively easy to understand its advantages and disadvantages.

While it is tempting to find a network correspondence with a single cell patch clamp, it might be difficult for at least two reasons: indirect network effects and speed. First, the advantage with single cell patch clamp is that there is a minimal impact on the remaining network since only one cell is modified. Here a whole population is modulated. Therefore a crucial question is how this modulation spreads through the network, how this spread changes the processing, and how this spread even will change the input that should be measured in the first place. Second, the main output of the classic voltage clamping is the amount of current that is required to compensate for various currents into a neuron. It is important that the compensation is complete, otherwise the injected current cannot be interpreted easily. This requires the regulator to be very fast. This is relatively easy with the membrane potential and currents for a patched cell, but not for a population of heterogeneously firing cells. It would be even more difficult to control for the firing of an individual neuron. This might explain why the controller needs some time to accumulate the spikes (tau=0.16s). This will unfortunately not be fast enough to follow the very fast transients caused by vibrissa stimulation (see Figure 8). Since the control feedback speed probably depends on the temporal resolution of the neuronal signal, it would be interesting to study what happens when the temporal resolution is increased by taking more and more neurons into account. In this case it may be possible to decrease the time constant such that fast network dynamics can be studied.

---

## [Author Response]

*1) The mechanisms underlying the light-mediated alternation or clamp of neuron excitability are highly dependent on various factors including neuron types expressing ChR2, their cellular and synaptic properties and interconnectivity. It was unclear how these factors may influence the speed and quality of clamping neuronal excitability*.

To address the concern about how variability of neural circuits and opsin expression profiles might affect features of activity other than the mean firing rate, we have clarified the Results section “Proportional-integral control of network firing” and the Discussion. Specifically, we make it clear that currently our method only controls the mean network firing level. We point out that even during successful control, higher-order characteristics of network activity, such as how the activity spreads through the network following stimulus application, unit-to-unit firing correlations, and firing synchrony remain uncontrolled. We point out that these characteristics are promising candidates for future extensions of our technique. Further, we emphasize that the optoclamp will almost certainly need to be used in combination with other methods in order to isolate mechanisms driving changes in network excitability or information processing that occur during the clamping period.

*2) One reviewer pointed out that it is unclear how the activity spreads through the network and how it may change the input. Finally, the reviewer pointed out that the speed of the system is a limiting factor in a fast control of network excitability. The question is whether the speed can be improved. In general, a more thorough discussion of the limitations of the techniques should be discussed. Below you will find the major and minor criticism of the two reviewers*.

We have made several changes to improve the clarity and thoroughness of our discussion of the technique’s general limitations. We also address the second reviewers’ concern of loop speed with an additional paragraph in the Discussion.

*Reviewer #1*:

*Major criticism*:

*By comparing sinusoidal light application to ChR2 expressing cell populations in comparison to rectangular or triangular light application, the authors observed differences in spike correlations and synchrony in some of the applied light protocols. The problem with this approach is that the resonance behavior of neurons is highly diverse and depends on the intrinsic membrane properties. Thus, any interpretations on the population behavior will depend on the nature of the cells expressing ChR2 and the percent of the contributing neuron types expressing CR2. Thus, the mentioned 'systematic modulation' of neuronal networks at the end of the subsection headed “Proportional-integral control of network firing” is very much dependent on various factors such as cell types, percent of neuron types expressing Chr2, brain area under investigation. This should be discussed in the manuscript*.

We thank the reviewer for making this point. Indeed, in its current form, the optoclamp is only capable of controlling the population firing rate. Higher order features of neural activity are not under active control and therefore are likely to be affected by the architecture and resonant dynamics of the circuit whose firing is being controlled. We were aware of this issue in the initial submission and attempted to address it in the Discussion:

Original: “Further, improved control algorithms might allow control over the heterogeneity of activity levels across cells, and temporal variance of firing activity (regular firing vs. bursting).”

We agree that this limitation needed a stronger explanation. To address this, we have changed the sentence:

Original: “Therefore, altering the temporal characteristics of excitatory stimulus waveforms permits systematic modulation of higher-order firing statistics during PI control of firing rate.”

to:

“Therefore, altering the temporal characteristics of excitatory stimulus waveforms resulted in remarkably different higher-order firing statistics while still enabling successful PI control of the mean firing rate. This emphasizes the fact that, in its current form, the optoclamp only controls population firing levels and leaves more complex features of neural activity unconstrained and subject to the influence of network connectivity, network dynamics, and the nature of the stimulus signal.” (This appears at the end of the subsection “Proportional-integral control of network firing.”)

Additionally, in the Discussion, we expand our explanation of this limitation using the following new paragraph:

“We demonstrated that optical waveforms with very different temporal characteristics could be used to successfully control population firing rate in-vitro […] such as the heterogeneity of activity levels across cells and/or temporal variance of firing activity (e.g. regular firing vs. bursting).”

*Minor criticism*:

*At the end of the first paragraph of the subsection headed “Multi-hour control of firing rates” I would propose to replace the wording 'network plasticity' by 'changes in neuronal network dynamics'*.

We agree that ‘network plasticity’ was too specific in this context and have made the requested change.

*Reviewer #2*:

*In this manuscript the authors propose a way to measure the input into a neuronal population by quantifying the optogenetic current which is required to keep those cells at a constant firing rate level. They use eNpHR3.0 to balance the lack of inhibition and ChR2 to counteract the lack of excitation in vitro and in vivo. With this method, the authors approximate the input to the neuronal population by the power of the yellow light minus the power of the blue light. Since the study controls for the neuronal firing, the clamp can be used to cancel slow components for several seconds. This was used to keep the neuronal firing rate constant in order to show that the synaptic homeostasis is not dependent on neuronal firing rate. This is a nice application of the method. Overall the paper is well written and the figures are clear. What is missing is a discussion and data about how this method differs from classic patch clamp and how one can deal with the differences. The patch clamp is an important reference since it is relatively easy to understand its advantages and disadvantages*.

We agree that the voltage clamp offers a nice point for comparison since it is the most widely applied use of feedback control in electrophysiology. Indeed, the Discussion provides a detailed comparison of our method with the voltage clamp to show their analogous features, and to highlight advantages and shortcomings of our technique**.** However, we see how the organization of these paragraphs was unclear, and have made changes to deal with this. In paragraph 4 of the Discussion, we compare the relative abilities of the voltage clamp and optoclamp to provide real-time readout of population and neural excitability, and detail the shortcomings of using voltage and optoclamping to perform these measurements. In paragraph 5, which has been expanded considerably, we state how experimentalists have dealt with imperfections in the voltage clamp technique to make it so powerful and reliable, and state how analogous compensating factors will be needed in to optoclamp to achieve the same result. In paragraph 6, we compare the analogous capabilities of voltage clamp and optoclamp for decoupling the membrane voltage or population firing rate, respectively, from other processes to which it would normally be causally dependent on.

However, we do stress that our technique and the voltage clamp serve entirely different purposes. Just as the reviewer states in the following paragraph, the voltage clamp is not capable of enforcing control over network dynamics. This is precisely what the optoclamp is designed to do. The firing activity measure obtained through electrical recording serves as a proxy for the firing of a much larger population of cells which are affected by optical input. So, while we agree that it is instructive to compare analogous properties of the two techniques, direct comparison is not meaningful since they are designed for completely different experimental needs.

*While it is tempting to find a network correspondence with a single cell patch clamp, it might be difficult for at least two reasons: indirect network effects and speed. First, the advantage with single cell patch clamp is that there is a minimal impact on the remaining network since only one cell is modified. Here a whole population is modulated. Therefore a crucial question is how this modulation spreads through the network, how this spread changes the processing, and how this spread even will change the input that should be measured in the first place*.

We agree with the reviewer’s point. We feel that we address this legitimate concern fairly extensively in Figures 4 and 6 as well as the Discussion. However, we agree that this explanation could have been made clearer. In Figure 4, we show how using different waveforms during closed loop PI control permits successful clamping of mean firing rates while producing very different correlation and synchrony structures in population activity (we have expanded our discussion of this limitation per the suggestion of Reviewer 1; see above). In Figure 6, we show how mean firing rate clamping can be achieved even during gross manipulation of network connectivity using different synaptic blockers, but unsurprisingly, the short time-scale features of population firing activity are greatly affected by these perturbations (Figure 6) even though mean firing levels are maintained. In the Discussion, we describe how the presence of uncontrolled features of network activity are analogous to uncontrolled variables during voltage clamping:

“One shortcoming [of the optoclamp] is that it is difficult to identify the origin of changes in network excitability that appear in optical control signals since they result from the aggregate effect of many different processes. However, parallel limitations also exist for the voltage clamp. For example, when using voltage clamp, it can be difficult to distinguish contributions from different neurotransmitter systems and intrinsic conductances or to deduce the relationship between synaptic currents measured at the recording site versus at the site of receptor binding.”

“Another issue with both optical clamping and the voltage clamp is that the control signal can be contaminated by non-ideal aspects of the recording or stimulation setup. For instance, opsin desensitization could affect the intensity of light required for the optoclamp to maintain firing levels. Likewise, space clamp and seal stability issues that arise during voltage clamp influence the amount of current required to hold a target membrane potential.”

In the original Discussion, we then went on to describe how these issues were tackled in the case of the voltage clamp and emphasize that any practical use of our technique will require analogous treatment. However, this section could have been clearer and more direct. To address this issue, we have modified the Discussion to emphasize that the optoclamp will almost certainly need to be used in combination with other methods in order to isolate mechanisms driving changes in network excitability or information processing that occur during the clamping period. Specifically, we added the following paragraph in the Discussion:

“In the case of the voltage clamp, these issues are partially compensated for by using secondary measurements of seal quality […] the optoclamp can be combined with existing pharmacological, genetic, or electrophysiological tools to isolate specific mechanisms that influence network excitability following clamping (Fong 2015).”

*Second, the main output of the classic voltage clamping is the amount of current that is required to compensate for various currents into a neuron. It is important that the compensation is complete, otherwise the injected current cannot be interpreted easily. This requires the regulator to be very fast. This is relatively easy with the membrane potential and currents for a patched cell, but not for a population of heterogeneously firing cells. It would be even more difficult to control for the firing of an individual neuron. This might explain why the controller needs some time to accumulate the spikes (tau=0.16s). This will unfortunately not be fast enough to follow the very fast transients caused by vibrissa stimulation (see*
Figure 8*). Since the control feedback speed probably depends on the temporal resolution of the neuronal signal, it would be interesting to study what happens when the temporal resolution is increased by taking more and more neurons into account. In this case it may be possible to decrease the time constant such that fast network dynamics can be studied*.

We agree with the reviewer. To address this concern, we have added the following text to the Discussion:

“Additionally, decreased feedback latency or the addition of predictive elements to the feedback loop may enable control over rapid sensory or motor events […] for which accurate input/output relationships can be deduced in situ using real-time system identification (Grosenick 2015).”